# Training Deep Spiking Neural Networks without Normalization

**Xinyu Shi** [1 2]  **Zhaofei Yu** [1 2]

## Abstract

The training of deep Spiking Neural Networks (SNNs) has traditionally relied on Batch Normalization (BN), which stabilizes input currents and gradients during training. However, BN is not a universal solution. It is unsuitable for variable-length tasks and scenarios with reduced batch size, constraining the development of deep SNNs, where removing BN typically causes the training to fail to converge. This dependence stems not from a fundamental necessity of BN but from the current lack of reasonable initialization methods for SNNs. This paper addresses this core limitation by proposing SpikeInit, a novel initialization framework for SNNs. By modeling the response curve and gradient of spiking layers, SpikeInit initializes the weights and shape parameters of surrogate gradients to maintain stable firing rates during forward propagation and stable gradient magnitudes during backpropagation. Extensive experiments demonstrate that deep SNNs with SpikeInit can be trained stably without normalization and achieve superior performance compared to their normalized counterparts under identical settings. Furthermore, we demonstrate the scalability of SpikeInit by successfully training an ultra-deep, 1000-layer SNN without normalization. Our work provides a foundational step toward large-scale normalization-free SNN, liberating SNN design from the constraints of normalization. Codes are available at: https://github.com/xyshi2000/SpikeInit

## 1. Introduction

The development of training methods for Spiking Neural Networks (SNNs), particularly Spatio-Temporal Backprop-agation (STBP) (Wu et al., 2018) and Surrogate Gradient (SG) learning (Neftci et al., 2019), has enabled the direct, end-to-end optimization of spiking models. Building on these advances, researchers have successfully adapted Batch Normalization (BN) (Ioffe & Szegedy, 2015) from Artificial Neural Networks (ANNs) to SNNs. It helps normalize the distribution of input currents and stabilize the gradient backpropagation, facilitating the training of deep and large-scale SNNs (Zheng et al., 2021), including Convolutional SNNs (Lee et al., 2020; Zheng et al., 2021; Fang et al., 2021a; Hu et al., 2024) and Spiking Vision Transformers (Zhou et al., 2023; Yao et al., 2023; Shi et al., 2024b; Zhou et al., 2024). Compared to other normalization methods, BN uses population statistics during inference and can be integrated into the preceding linear transformations, thereby avoiding the floating-point multiplication and division operations required to compute statistics. This preserves the spike-driven nature of SNNs, making BN particularly suitable for current deep SNN architectures.

However, Batch Normalization is not a universal solution. While it has proven effective for fixed-size visual tasks, BN is generally unsuitable for natural language processing tasks due to the variability in input sequence lengths. Specifically, ANNs typically rely on instance-specific techniques such as layer normalization or RMSNorm. Unfortunately, applying these methods to SNNs introduces normalization operations during inference, which require floating-point multiplications that fundamentally disrupt the energy-efficient, spike-driven nature of SNNs. Furthermore, BN requires a sufficiently large batch size to compute accurate statistics. This becomes a significant limitation in memory-intensive scenarios, such as high-resolution vision tasks or large-scale Vision Transformers, where reducing batch size is often necessary. These inherent limitations constrain the development of deep SNNs that rely on BN for convergence.

Recent studies on ANNs demonstrate that explicit normalization is not strictly essential (Zhang et al., 2019). Deep ANNs with proper initialization can achieve performance comparable to or surpassing their normalized counterparts (Brock et al., 2021; De & Smith, 2020; Bachlechner et al., 2021). In contrast, removing BN from deep SNNs typically causes training to fail immediately. However, recent research indicates that this instability does not stem from the inherent necessity of normalization for SNNs, but rather

[1]Institute for Artificial Intelligence, Peking University [2]Beijing Key Laboratory of Brain-inspired Spiking Large Models, School of Computer Science, Peking University. Correspondence to: Zhaofei Yu <yuzf12@pku.edu.cn>.

*Proceedings of the 43rd International Conference on Machine Learning*, Seoul, South Korea. PMLR 306, 2026. Copyright 2026 by the author(s).

from the current lack of reasonable initialization methods for SNNs (Micheli et al., 2025). Normalization effectively mitigates this issue by enforcing signal stability, thereby obscuring the fundamental problem in initialization.

Despite this, many deep SNNs still employ initialization methods originally designed for ANNs, such as Xavier (Glorot & Bengio, 2010) and Kaiming initialization (He et al., 2015). These methods derive scaling factors based on the properties of continuous activation functions, such as ReLU or tanh, without accounting for the binary activation and temporal dynamics unique to spiking neurons. Directly applying these ANN initialization methods to SNNs often leads to spike vanishing in deep layers, rendering them inactive and untrainable.

Previous studies have modeled the activation characteristics of spiking neurons and proposed several weight initialization methods for SNNs, enabling the training of certain shallow networks without normalization (Rossbroich et al., 2022; Ding et al., 2025; Micheli et al., 2025). However, the temporal dynamic responses of spiking neurons modeled by these methods are not sufficiently precise. The approximations used, such as mean input current or considering the first time step only, introduce deviation from the actual temporal dynamics of spiking neurons, preventing them from scaling to deep, large-scale SNNs. Moreover, their performance still lags behind the normalized counterparts.

In this paper, we model the response curve of typical spiking layers by sufficiently accounting for the temporal dynamics, and propose a weight initialization method that maintains stable firing rates across layers during forward propagation. Furthermore, we formulate the gradient of each layer and propose an initialization method for the shape parameter of the surrogate gradient function, ensuring stable gradient magnitudes across layers during backpropagation. Combining these contributions, we propose SpikeInit, an initialization framework for SNNs without normalization. Extensive experimental results demonstrate that with SpikeInit, we can stably train deep SNNs without normalization, outperforming normalized counterparts under identical hyperparameter settings. Moreover, we successfully train a 1000-layer ultra-deep normalization-free SNN using SpikeInit, demonstrating its superior scalability. Our main contributions are summarized as follows:

- We model the response curve of typical spiking layers and propose a weight initialization method that maintains stable firing rates across layers in forward propagation.

- We formulate the gradient of spiking layers and propose an initialization method for the shape parameter of surrogate gradients that maintains stable gradient magnitudes across layers during backpropagation.

- Combining these, we propose SpikeInit, an initialization framework for SNNs. Experimental results show that SpikeInit can stably train deep SNNs without normalization and outperform normalized counterparts.

## 2. Related Work

**Batch Normalization in SNNs.** One technical route to incorporating BN into SNNs focuses on handling the additional temporal dimension. Batch Normalization Through Time (BNTT) (Kim & Panda, 2021) decouples the scaling parameters for each time step, effectively allowing the network to learn a time-variant threshold. Threshold-dependent Batch Normalization (tdBN) (Zheng et al., 2021) folds the temporal and batch dimensions to compute global statistics and explicitly incorporates the threshold into the affine transformation, theoretically ensuring "Block Dynamical Isometry". Temporal Effective Batch Normalization (TEBN) (Deng et al., 2022) maintains shared statistics and introduces a lightweight, time-dependent weighting mechanism to rescale inputs. Temporal Accumulated Batch Normalization (TAB) (Jiang et al., 2024) computes statistics using a sliding window accumulation rather than instantaneous or global averages.

A distinct line of research finds that normalizing presynaptic currents is insufficient because the membrane potential update introduces disturbance before activation. Potential Normalization (PSP-BN) (Ikegawa et al., 2022) addressed this by normalizing using the second raw moment rather than variance to preserve the sign and the absolute magnitude structure of the membrane potential. Membrane Potential Batch Normalization (MPBN) (Guo et al., 2023) applies normalization directly to the membrane potential.

**Weight initialization for SNNs.** Early efforts to initialize SNNs largely rely on empirical settings for the magnitude of weights (Lee et al., 2016; Wu et al., 2018; Zenke & Vogels, 2021). However, these methods lack a solid theoretical foundation and scalability. Fluctuation-Driven Initialization (Rossbroich et al., 2022) introduces a biologically inspired approach, deriving weight scales to maintain a "balanced state" where firing is driven by membrane potential fluctuations. However, this method still depends on the empirical tuning of the hyperparameter $\xi$. Another study (Ding et al., 2025) analyzes the asymptotic response curve of spiking neurons under mean-driven input, and propose a weight initialization method to operate within the linear "slant asymptote" region of the surrogate gradient function. Nevertheless, this approach overlooks the temporal randomness of inputs, leading to an overestimation of variance. A more recent work (Micheli et al., 2025) scales synaptic weights inversely to the firing rate to maintain constant signal variance across layers. However, the analysis is limited to the first time step and lacks a comprehensive

temporal dynamic model of the membrane potential's response to input. In summary, while progress has been made, current SNN initialization methods still face challenges in theoretical rigor, scalability, and fully accounting for temporal dynamics.

## 3. Preliminary

### 3.1. Spiking Neuron Model

Similar to (Ding et al., 2025), we use the following unified discrete-time model to describe the neural dynamics of spiking neurons:

$$v[t] = \kappa u[t] + \lambda x[t], \tag{1}$$
$$s[t] = H(v[t] - v_{\text{th}}), \tag{2}$$
$$u[t+1] = (1 - s[t])v[t]. \tag{3}$$

Here $x[t]$ denotes the input current of the neuron at timestep $t$. $u[t]$ and $v[t]$ denote the membrane potential of the neuron before and after charging at time-step $t$. $\kappa$ is the decay factor of the membrane potential. $\lambda$ is the scaling factor of the input current. $s[t] \in \{0, 1\}$ is the output of the neuron at time-step $t$, where $s[t] = 1$ denotes the neuron fires a spike. $H(\cdot)$ is the Heaviside step function. If the membrane potential after charging $v[t]$ exceeds the firing threshold $v_{\text{th}}$, the neuron fires a spike, and the membrane potential after firing $u[t+1]$ will reset to the resting potential. Since the rest potential is typically set to zero, we omit it in subsequent analyses.

By setting different values for $\kappa$ and $\lambda$, the charging equation Eq. (1) can describe different neuron models. For example, $\kappa = \lambda = 1$ formulates the integrate-and-fire (IF) model. $0 < \kappa < 1$ and $\lambda = 1 - \kappa$ formulates the leaky integrate-and-fire (LIF) model with input decay (Fang et al., 2021a;b). $0 < \kappa < 1$ and $\lambda = 1$ formulates the LIF model without input decay (Wu et al., 2018; Zheng et al., 2021; Hu et al., 2024). In this paper, we use $\kappa = 0.5$, $\lambda = 0.5$, $v_{\text{th}} = 0.5$ by default.

### 3.2. Surrogate Gradient

We calculate the gradients using spatio-temporal backpropagation (STBP) (Wu et al., 2018) as

$$\frac{\partial \mathcal{L}}{\partial W} = \sum_t \frac{\partial \mathcal{L}}{\partial s[t]} \frac{\partial s[t]}{\partial v[t]} \frac{\partial v[t]}{\partial x[t]} \frac{\partial x[t]}{\partial W}. \tag{4}$$

Here the gradient of spikes $s[t]$ with respect to the membrane potential $v[t]$ is the gradient of Heaviside step function $H(\cdot)$

$$\frac{\partial s[t]}{\partial v[t]} = \frac{\partial H(v[t] - v_{\text{th}})}{\partial v[t]}, \tag{5}$$

which is non-differentiable. To solve this problem, surrogate gradient (SG) is typically employed to replace the

gradient of Heaviside function. It utilizes a differentiable surrogate function $\Theta(\cdot; \alpha)$ to approximate the Heaviside function $H(\cdot)$

$$\frac{\partial s[t]}{\partial v[t]} \approx \frac{\partial \Theta(\alpha(v[t] - v_{\text{th}}))}{\partial v[t]}, \tag{6}$$

where $\alpha$ is the shape parameter that controls the smoothness of the gradient. Common surrogate gradients include triangle (Wu et al., 2018), arc-tangent (Fang et al., 2021a), exponential (Neftci et al., 2019), etc. In this paper, we use the exponential surrogate gradient by default, i.e., $\partial \Theta / \partial v = \alpha e^{-2\alpha|v - v_{\text{th}}|}$.

## 4. Weight Initialization for Spiking Neural Networks

A typical ANN layer can be formulated as

$$y^{(l)} = W^{(l)} x^{(l)} + b^{(l)}, \tag{7}$$
$$x^{(l+1)} = f(y^{(l)}), \tag{8}$$

where $f(\cdot)$ is the nonlinear activation function, for example, $\text{ReLU}(x) = \max(x, 0)$. It consists of two sub-processes: the linear transformation and the nonlinear activation, formulated by Eq. (7) and Eq. (8), respectively. The core principle of ANN weight initialization methods (Glorot & Bengio, 2010; He et al., 2015) is to maintain magnitude stability in activation values across network layers, i.e., $\text{Var}(x^{(l+1)}) = \text{Var}(x^{(l)})$, by appropriately initializing weights. In contrast, a typical SNN layer takes the form of

$$x^{(l)}[t] = \sum_i w_i^{(l)} s_i^{(l)}[t], \tag{9}$$

$$v^{(l)}[t] = \kappa u^{(l)}[t] + \lambda x^{(l)}[t], \tag{10}$$

$$s^{(l+1)}[t] = H(v^{(l)}[t] - v_{\text{th}}), \tag{11}$$

$$u^{(l)}[t+1] = (1 - s^{(l+1)}[t])v^{(l)}[t]. \tag{12}$$

It differs from ANNs in three key aspects: 1) It has the additional charging and resetting processes as formulated in Eq. (10) and Eq. (12). 2) It has an additional temporal dimension with corresponding temporal dynamics. 3) The activation values in SNNs are discrete binary spikes, whereas those in ANNs are continuous values. These differences make variance analysis in ANNs unsuitable for SNNs. Consequently, weight initialization methods designed for ANNs are also not directly applicable to SNNs.

For a randomly initialized SNN, we assume that the spiking activity in each layer follows a Poisson process. Therefore, spike outputs are subject to a Bernoulli distribution, i.e.,

$$s^{(l)}[t] \sim \text{Bernoulli}(p^{(l)}), \tag{13}$$

where $p^{(l)}$ represents the firing rate of this layer. Since the variance of the Bernoulli distribution is $p(1 - p)$, the prob-

lem is transformed into maintaining the firing rate stability across layers, i.e., $p^{(l+1)} = p^{(l)}$.

## 4.1. Weight Initialization based on Firing Rate Response Curve

To formulate the relation between $p^{(l)}$ and $p^{(l+1)}$, we first integrate the charging and resetting into a membrane potential update process. Then we derive this relation stepwise by analyzing how the input spike firing rate $p^{(l)}$ influences the input current $x$, then the membrane potential $v$, and finally the output spike firing rate $p^{(l+1)}$. The analysis separately considers the linear transformation, membrane potential update, and activation processes.

First, consider the linear transformation. Assume that the weights are independently drawn from the same distribution with mean 0 and variance $(\sigma_w^{(l)})^2$ at initialization. Furthermore, we assume that the input spikes are independently drawn from the same Bernoulli distribution with parameter $p^{(l)}$. Finally, weights and input spikes are assumed to be mutually independent. Therefore, according to Eq. (9) and the central limit theorem, the input current is approximately subject to a normal distribution

$$x^{(l)}[t] \sim \mathcal{N}(0, n_{\text{in}}^{(l)} p^{(l)} (\sigma_w^{(l)})^2), \qquad (14)$$

where $n_{\text{in}}^{(l)}$ denotes the number of input neurons (fan-in). Since weights are typically scaled by $\sqrt{n}$, i.e., $\sigma_w^{(l)} \propto 1/\sqrt{n_{\text{in}}^{(l)}}$, we abbreviate $\sigma_w^2 = n_{\text{in}}^{(l)} (\sigma_w^{(l)})^2$ in the following. Second, consider the activation. The output firing rate $p^{(l+1)}$ can be calculated using the distribution of membrane potential

$$p^{(l+1)} = \int_{v_{\text{th}}}^{\infty} f_v(u) du = 1 - F_v(v_{\text{th}}), \qquad (15)$$

where $f_v(\cdot)$ and $F_v(\cdot)$ are the probability density function (PDF) and cumulative distribution function (CDF) of the membrane potential, respectively.

The main challenge is the membrane potential update process. Considering only the membrane potential after charging $v$, the temporal dynamics of the membrane potential can be formulated using the following threshold auto-regressive (TAR) process

$$v[t+1] = \begin{cases} \kappa v[t] + x[t+1] & \text{if } v[t] < v_{\text{th}} \\ x[t+1] & \text{if } v[t] \geq v_{\text{th}}. \end{cases} \qquad (16)$$

Here we omit $\lambda$, since it can be merged into the variance of $x$. Unfortunately, the PDF of the membrane potential under such a TAR process does not maintain a simple closed form, since the reset condition depends on the current value of the membrane potential. Instead, the distribution is defined by

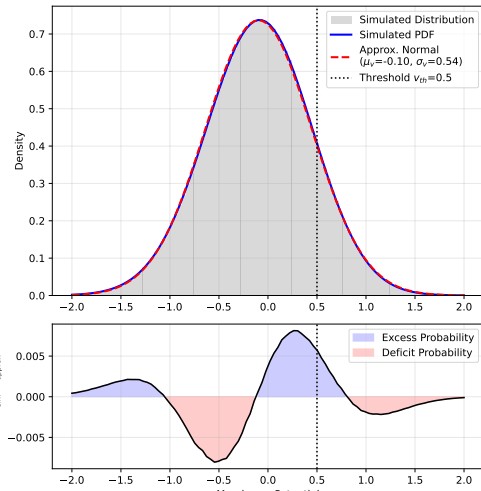

*Figure 1.* **Top**: Distribution of membrane potential under $\kappa = 0.5$, $v_{\text{th}} = 0.5$, $\sigma_x = 0.5$. The blue solid line represents the PDF obtained through Monte Carlo simulation. The red dashed line represents the PDF of the approximate normal distribution. **Bottom**: Error of the approximate PDF.

a recursive integral equation

$$f_{t+1}(v) = \frac{1}{\sigma_x} \phi \left( \frac{v}{\sigma_x} \right) \int_{v_{\text{th}}}^{\infty} f_t(u) \, du$$
$$+ \int_{-\infty}^{v_{\text{th}}} f_t(u) \frac{1}{\sigma_x} \phi \left( \frac{v - \kappa u}{\sigma_x} \right) \, du, \qquad (17)$$

where $\sigma_x$ denotes the deviation of the input current $x$. $f_t(\cdot)$ and $f_{t+1}(\cdot)$ denote the PDF of $v[t]$ and $v[t+1]$, respectively. The first term represents the reset condition, where the distribution of membrane potential adopts the normal distribution of input current. The second term represents the accumulation condition, where the distribution of membrane potential is a convolution of input with the truncated distribution of the previous step.

While Eq. (17) does not have a simple closed form, we find that the membrane potential closely approximates the normal distribution when the decay factor $\kappa$ is small. Fig. 1 shows the actual PDF of the membrane potential obtained through Monte Carlo simulations and its approximate normal distribution under $\kappa = 0.5$. The two distributions are highly similar, with a maximum error of less than 0.01 in the PDF. Therefore, we can approximate the probability density function of the membrane potential as a normal distribution. Detailed error analysis of the approximation and alternative method when $\kappa$ is close to 1 can be found in Appendix C.

Assuming the membrane potential converges to a stationary distribution approximating a normal distribution, we can calculate the expectation and variance of this approximate normal distribution following Eq. (16). Specifically, the expectation and variance of the stationary distribution are formulated by the following theorem

**Algorithm 1** Firing Rate Response

**Input:** $p^{(l)}, \sigma_w^2, \kappa, \lambda, v_{\text{th}}$
**Output:** $p^{(l+1)}$
 1: $\sigma_x \leftarrow \sqrt{\lambda p^{(l)} \sigma_w^2}$
 2: Calculate $\mu_v, \sigma_v$ by solving Eq. (18) and Eq. (19) given $\sigma_x, \kappa$, and $v_{\text{th}}$
 3: $p^{(l+1)} \leftarrow 1 - \Phi\left(\frac{v_{\text{th}} - \mu_v}{\sigma_v}\right)$

---

**Algorithm 2** Binary Search $\sigma^*$

**Input:** $p_{\text{init}}, \kappa, \lambda, v_{\text{th}}, \sigma_{\max}$
**Output:** $\sigma^*$
 1: $\sigma_{\text{low}} \leftarrow 0, \sigma_{\text{high}} \leftarrow \sigma_{\max}$
 2: **while** not converge **do**
 3: $\quad \sigma^* \leftarrow (\sigma_{\text{high}} + \sigma_{\text{low}})/2$
 4: $\quad$ Calculate $p_{\text{response}}$ by Algorithm 1 given $\sigma^*$
 5: $\quad$ **if** $|p_{\text{response}} - p_{\text{init}}| < \epsilon$ **then**
 6: $\quad\quad$ **break**
 7: $\quad$ **else if** $p_{\text{response}} < p_{\text{init}}$ **then**
 8: $\quad\quad \sigma_{\text{low}} \leftarrow \sigma^*$
 9: $\quad$ **else**
10: $\quad\quad \sigma_{\text{high}} \leftarrow \sigma^*$
11: $\quad$ **end if**
12: **end while**

---

**Theorem 4.1.** *Let $\{x[t]\}_{t \geq 0}$ be a sequence of independent, identically distributed random variables such that $x[t] \sim \mathcal{N}(0, \sigma_x^2)$. Let $v[t]$ be a stochastic process defined by the recurrence in Eq. (16). Assume that the stationary distribution of $v[t]$ exists and can be approximated by a Normal distribution $v \sim \mathcal{N}(\mu_v, \sigma_v^2)$. Then, the parameters $\mu_v$ and $\sigma_v$ are the solutions to the following system of coupled nonlinear equations:*

$$\mu_v(1 - \kappa\Phi(a)) = -\kappa\sigma_v\phi(a), \tag{18}$$

$$\mu_v^2 + \sigma_v^2 = \sigma_x^2 + \kappa^2((\mu_v^2 + \sigma_v^2)\Phi(a) - \sigma_v(\mu_v + v_{\text{th}})\phi(a)), \tag{19}$$

*where $a = (v_{\text{th}} - \mu_v)/\sigma_v$ is the standardized threshold, and $\phi(\cdot), \Phi(\cdot)$ are the PDF and CDF of the standard normal distribution, respectively.*

A detailed proof can be found in Appendix A.1. Although the system of nonlinear equations in Theorem 4.1 has no analytical solution, it can be easily solved using numerical methods[1]. Based on Theorem 4.1, we can calculate $p^{(l+1)}$ from $p^{(l)}$ using algorithm 1, thereby deriving the firing rate response curve, denoted as $q(\cdot)$, i.e., $p^{(l+1)} = q(p^{(l)})$.

As shown in Fig. 2, the firing rate response curve is nonlinear, which means we cannot guarantee $p^{(l+1)} = q(p^{(l)}) =$

---

[1]We use the fsolve function to solve the system of nonlinear equations in implementation

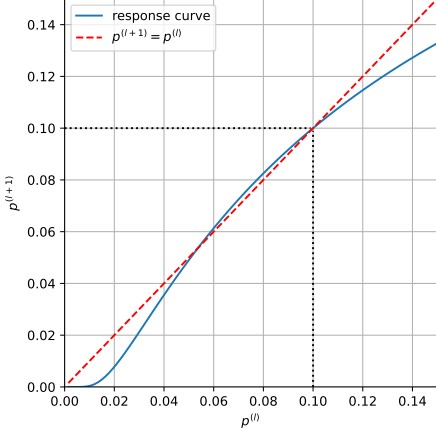

*Figure 2.* Firing rate response curve under $\kappa = 0.5$, $v_{\text{th}} = 0.5$, $p_{\text{init}} = 0.1$, and $\sigma^*$ calculated by Algorithm 2

$p^{(l)}$ for arbitrary $p^{(l)}$. However, for a specified target initial firing rate $p_{\text{init}}$, we can adjust $\sigma_w$ to obtain a response curve satisfying $p_{\text{init}} = q(p_{\text{init}})$. Note that $q(p_{\text{init}})$ increases monotonically with $\sigma_w$, this can be efficiently implemented by binary search. In summary, given $p_{\text{init}}$ and hyperparameters of the LIF model, we can search for the optimal $\sigma^*$ using Algorithm 2, and initialize weights in each layer as

$$w^{(l)} \sim \mathcal{N}(0, (\sigma^*)^2 / n_{\text{in}}^{(l)}). \tag{20}$$

This ensures that the network forward propagates at a stable firing rate $p_{\text{init}}$ in its initial state. In this paper, we set $p_{\text{init}} = 0.1$ by default.

The above analysis considers only the standard hidden layers. Initialization for special layers, including the first layer that converts the input to spike trains with target initial firing rate $p_{\text{init}}$ and output layers in residual blocks, is detailed in Appendix B.

## 5. Initialization of Surrogate Gradient

In Section 4, we focus on forward propagation. It is also important to maintain magnitude stability in back-propagated gradients, i.e., $\text{Var}(\partial\mathcal{L}/\partial s^{(l)}) = \text{Var}(\partial\mathcal{L}/\partial s^{(l+1)})$. In ANNs, the variances of both activations and their gradients are affected by the variance of weights, which follows a simple $n_{\text{in}}/n_{\text{out}}$ ratio (Glorot & Bengio, 2010; He et al., 2015) based on forward or backward propagation constraints. He et al. (2015) find that it is sufficient to initialize weights by following the constraint of either the forward or the backward propagation case.

However, in SNNs, maintaining stable variance in back-propagated gradients cannot be achieved solely through weight initialization. This is because we employ the surrogate gradient (SG) for the non-differentiable Heaviside function. The shape parameter $\alpha$ in SG influences the gradi-

ent distribution and thus affects its variance. Therefore, to preserve gradient variance stability during backward propagation, it is necessary to appropriately set the shape parameter $\alpha$ in the surrogate gradient function.

## 5.1. Initialization of Shape Parameter

We formulate the gradient of each layer as

$$
\begin{aligned}
\frac{\partial \mathcal{L}}{\partial \mathrm{s}^{(l)}[t]} &= \sum_{i=t}^{T} \frac{\partial \mathcal{L}}{\partial \mathrm{s}^{(l+1)}[i]} \frac{\partial \mathrm{s}^{(l+1)}[i]}{\partial \mathrm{v}^{(l)}[i]} \frac{\partial \mathrm{v}^{(l)}[i]}{\partial \mathrm{x}^{(l)}[t]} \frac{\partial \mathrm{x}^{(l)}[t]}{\partial \mathrm{s}^{(l)}[t]} \\
&= \sum_{i=t}^{T} \frac{\partial \mathcal{L}}{\partial \mathrm{s}^{(l+1)}[i]} \frac{\partial \mathrm{s}^{(l+1)}[i]}{\partial \mathrm{v}^{(l)}[i]} \frac{\partial \mathrm{v}^{(l)}[i]}{\partial \mathrm{x}^{(l)}[t]} \mathrm{W}^{(l)}.
\end{aligned}
\tag{21}
$$

Since initialized weights have a mean of 0, the expectation of the gradient $E\left[\partial \mathcal{L}/\partial \mathrm{s}^{(l)}[t]\right] = 0$. However, it is difficult to precisely calculate the variance $\mathrm{Var}\left(\partial \mathcal{L}/\partial \mathrm{s}^{(l)}[t]\right)$, since the gradient terms $\partial \mathrm{s}^{(l+1)}[i]/\partial \mathrm{v}^{(l)}[i]$ and $\partial \mathrm{v}^{(l)}[i]/\partial \mathrm{x}^{(l)}[t]$ depends on the same membrane potential $v^{(l)}[i]$. Nevertheless, based on the stationary distribution of membrane potential and the assumption that spike outputs are subject to a Bernoulli distribution as used in Sec. 4, the variance of the gradient can be simplified and formulated as the following theorem

**Theorem 5.1.** *Let $v^{(l)}[t]$ be a stochastic process defined by the recurrence in Eq. (16). Assume that the reset probability (firing rate) is $p$, and the stationary distribution of $v^{(l)}[t]$ exists and can be approximated by a Normal distribution $v \sim \mathcal{N}(\mu_v, \sigma_v^2)$. Let the spike output be $s^{(l)}[t] = H(v^{(l)}[t] - v_{\mathrm{th}})$, where the derivative of the activation function is given by the surrogate gradient $\partial H(v - v_{\mathrm{th}})/\partial v \approx \partial \Theta(v; \alpha)/\partial v = \Theta'(v; \alpha)$. Assume the incoming gradients $\partial \mathcal{L}/\partial \mathrm{s}^{(l+1)}[t]$ are independent, zero-mean random variables with variance $\sigma_g^2$. Then, the variance of the total gradient of the loss with respect to the spike inputs is approximately given by*

$$
\mathrm{Var}\left(\frac{\partial \mathcal{L}}{\partial s^{(l)}[t]}\right) \approx \frac{n_{\mathrm{out}}^{(l)}(\sigma_w^{(l)})^2 \lambda^2 \sigma_g^2}{1 - \kappa^2(1-p)} \cdot \mathcal{M}_2, \tag{22}
$$

*where $n_{\mathrm{out}}^{(l)}$ denotes the number of output neurons (fan-out). $\mathcal{M}_2 = E[(\Theta'(v - v_{\mathrm{th}}; \alpha))^2]$ is the expectation of the squared derivative of the activation function under the steady-state distribution*

$$
\mathcal{M}_2 = \int_{-\infty}^{+\infty} (\Theta'(v - v_{\mathrm{th}}; \alpha))^2 \phi\left(\frac{v - \mu_v}{\sigma_v}\right) dv, \tag{23}
$$

*where $\phi(\cdot)$ is the PDF of the standard normal distribution.*

A detailed proof can be found in Appendix A.2. According to Theorem 5.1, the constraint of gradient magnitudes $\mathrm{Var}(\partial \mathcal{L}/\partial \mathrm{s}^{(l)}) = \mathrm{Var}(\partial \mathcal{L}/\partial \mathrm{s}^{(l+1)})$ equals to

$$
\mathcal{M}_2 = \frac{1 - \kappa^2(1-p)}{n_{\mathrm{out}}^{(l)}(\sigma_w^{(l)})^2 \lambda^2}. \tag{24}
$$

Specifically, the expectation of the squared derivative of the exponential surrogate function under the steady-state distribution can be formulated as

$$
\begin{aligned}
\mathcal{M}_2 &= \alpha^2 e^{-4\alpha(\mu_v - v_{\mathrm{th}}) + 8\alpha^2 \sigma_v^2} \Phi\left(\frac{v_{\mathrm{th}} - \mu_v - 4\alpha \sigma_v^2}{\sigma_v}\right) \\
&\quad + \alpha^2 e^{4\alpha(\mu_v - v_{\mathrm{th}}) + 8\alpha^2 \sigma_v^2} \Phi\left(\frac{\mu_v - v_{\mathrm{th}} - 4\alpha \sigma_v^2}{\sigma_v}\right).
\end{aligned}
\tag{25}
$$

The derivation can be found in Appendix A.3. Although Eq. (24) has no analytical solution, it can be solved using numerical methods[2]. Therefore, we calculate the initial $\alpha^*$ by solving Eq. (24). This ensures that the network maintains a stable gradient magnitude across layers during backpropagation.

## 5.2. Adaptive Surrogate Gradient

While the initialization of the shape parameter $\alpha$ in SG ensures stable gradient magnitudes at the initial state, a fixed shape parameter may not consistently match the dynamic distribution of membrane potential during training, as this distribution evolves with weight updates. Normalization layers help stabilize the membrane potential distribution by standardizing either the input current or the membrane potential itself. The membrane potential with normalization (Guo et al., 2023) can be formulated as

$$
\tilde{v} = \gamma \frac{v - \hat{\mu}_v}{\hat{\sigma}_v} + \beta, \tag{26}
$$

where $\gamma$ and $\beta$ are affine parameters, $\hat{\mu}_v$ and $\hat{\sigma}_v$ are the mean and standard deviation of the mini-batch during training and population statistics during inference. The surrogate function becomes

$$
\begin{aligned}
\Theta(\alpha(\tilde{v} - v_{\mathrm{th}})) &= \Theta\left(\alpha\left(\gamma \frac{v - \hat{\mu}_v}{\hat{\sigma}_v} + \beta - v_{\mathrm{th}}\right)\right) \\
&= \Theta\left(\frac{\alpha \gamma}{\hat{\sigma}_v}\left(v - \tilde{v_{\mathrm{th}}}\right)\right),
\end{aligned}
\tag{27}
$$

where $\tilde{v_{\mathrm{th}}} = \hat{\mu}_v + \sigma_v(v_{\mathrm{th}} - \beta)/\gamma$ is the reparameterized threshold during inference.

Since we do not incorporate normalization in the forward propagation, we introduce an adaptive shape parameter to achieve a similar stabilizing effect. Specifically, we set $\alpha = \alpha^* \cdot \sigma_v/\sigma_{v,i}$, where $\sigma_v$ is the standard deviation of the initial distribution of membrane potential calculated by Theorem 4.1, $\sigma_{v,i}$ is the standard deviation of membrane potential of sample $i$. Note that the surrogate gradients are only computed during backpropagation, therefore the adaptive shape parameter does not introduce additional computational overhead during inference. Similarly, we also

---

[2]We use the $\mathrm{brentq}$ function to solve the nonlinear equation in implementation

*Table 1.* Results on CIFAR-10 with different architectures. - denotes the training fails to converge.

| Method | Normalization Free | Accuracy (%) | | | |
|---|---|---|---|---|---|
| | | VGG-11 | VGG-16 | ResNet-18 | ResNet-34 |
| fluctuation-driven (Rossbroich et al., 2022) | ✓ | - | - | 89.15 | 90.96 |
| Ding et al. (2025) initialization | ✓ | - | - | 89.57 | 91.16 |
| Micheli et al. (2025) initialization | ✓ | - | - | 90.50 | 91.83 |
| BNTT (Kim & Panda, 2021) | ✗ | 93.32 | 94.49 | 94.63 | 95.38 |
| tdBN (Zheng et al., 2021) | ✗ | 93.56 | 92.86 | 95.37 | 95.86 |
| TEBN (Duan et al., 2022) | ✗ | 94.99 | 94.58 | 95.33 | 95.87 |
| MPBN (Guo et al., 2023) | ✗ | 94.62 | 95.31 | 94.64 | 95.47 |
| **SpikeInit (Ours)** | ✓ | **95.29** | **95.77** | **95.53** | **96.20** |

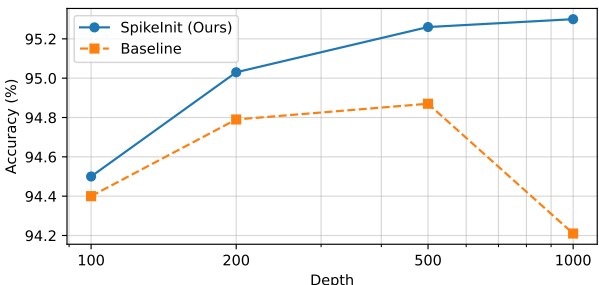

*Figure 3.* Comparison of SpikeInit and baseline MS-ResNet with tdBN as network depth increases. Results show that SpikeInit supports training ultra-deep SNNs, with accuracy continuously improving as the network depth increases.

apply affine parameters to the membrane potential, which can be integrated into the threshold during inference, thus also avoiding extra computations.

## 6. Experiments

In this section, we conduct experiments on different tasks using various architectures to evaluate the effectiveness, scalability, and generalizability of SpikeInit. We also perform an ablation study on key components of SpikeInit. Experimental settings are detailed in Appendix E. Further analysis regarding firing rates and gradients at initialization, as well as training stability and convergence, can be found in Appendix F and G.

### 6.1. Training Deep SNNs

One of the key advantages of Batch Normalization for SNNs is its scalability for training very deep SNNs (Hu et al., 2024). In this section, we investigate whether this scalability can be achieved solely through proper initialization. We conduct experiments on CIFAR-10 using MS-ResNet with increasing depth. For comparison, the same baseline architectures are trained with tdBN (Zheng et al., 2021), which is employed in the original MS-ResNet implemen-

*Table 2.* Results on ImageNet with ResNet-34.

| Method | Normalization Free | T | Accuracy (%) |
|---|---|---|---|
| tdBN (Zheng et al., 2021) | ✗ | 6 | 63.72 |
| MPBN (Guo et al., 2023) | ✗ | 4 | 64.71 |
| TAB (Jiang et al., 2024) | ✗ | 4 | 67.78 |
| TEBN (Duan et al., 2022) | ✗ | 4 | 68.28 |
| **SpikeInit (Ours)** | ✓ | **4** | **68.81** |
| | | **6** | **69.76** |

tation. All hyperparameter settings remain identical across experiments. The experimental results are shown in Fig. 3. The baseline approach exhibited performance degradation when the network depth reached 1000 layers, with its accuracy at this depth even falling below that achieved with 500 layers. In contrast, the accuracy of the proposed SpikeInit method shows continuous improvement as network depth increases, demonstrating its superior scalability for deep SNN architectures. Furthermore, SpikeInit consistently outperforms the baseline across different depths, demonstrating its effectiveness.

### 6.2. Image Classification

In this section, we evaluate the effectiveness of SpikeInit in image classification tasks. We first conduct experiments on CIFAR-10 using various network architectures, and compare with the state-of-the-art SNN initialization methods (Rossbroich et al., 2022; Ding et al., 2025; Micheli et al., 2025) and batch normalization adaptations for SNNs (Kim & Panda, 2021; Zheng et al., 2021; Duan et al., 2022). To ensure robustness, each experiment is repeated three times with different random seeds, and the average accuracy is reported. All hyperparameter settings are kept identical across experiments. The experimental results are summarized in Tab. 1. We observe that existing SNN initialization methods fail to converge on VGG-based architectures. This issue arises because the temporal dynamics modeled by these methods deviate from actual neuronal behavior, resulting in

*Table 3.* Results on IWSLT17 German-English with spiking transformer 4-256.

| Description | Normalization Free | BLEU |
|---|---|---|
| BatchNorm (tdBN) (Zheng et al., 2021) | ✗ | 28.14 |
| LayerNorm | ✗ | 29.50 |
| **SpikeInit (Ours)** | ✓ | **29.98** |

*Table 4.* Ablation study. - denotes the training fails to converge.

| Method | Accuracy (%) | |
|---|---|---|
| | VGG-11 | ResNet-18 |
| SpikeInit | 95.29 | 95.53 |
| w/o weight initialization | - | - |
| w/o surrogate gradient initialization | 78.96 | 91.05 |
| w/o adaptive surrogate gradient | 94.43 | 94.91 |

excessively small initial weight magnitudes. As a result, the spikes vanish after propagating through several layers, leading to training failure. Additionally, when applied to ResNet architectures, these methods achieve significantly lower accuracy compared to the normalized counterparts. In contrast, SpikeInit enables stable training across different network architectures and outperforms state-of-the-art BN adaptations for SNNs. Specifically, SpikeInit achieves 95.77% accuracy with VGG-16, surpassing the best-performing method MPBN by 0.46%. Furthermore, SpikeInit achieves 96.20% accuracy with MS ResNet-34, outperforming TEBN by 0.33%. These results demonstrate that normalization is not essential for training deep SNNs. With proper initialization, SNNs can perform competitively without normalization.

To further evaluate the scalability of SpikeInit on larger datasets, we conduct experiments on ImageNet using the MS ResNet-34 architecture. Since existing SNN initialization methods have not scaled to ImageNet classification, we compare our method with state-of-the-art BN adaptations for SNNs with ResNet-34, using the original results reported in their respective works. Since different methods employ varying time steps, we conduct experiments under multiple time-step settings to ensure a fair comparison. The results are listed in Tab. 2. SpikeInit achieves 68.81% accuracy under 4 time steps, outperforming TEBN by 0.53%, and reaches 69.76% accuracy under 6 time steps. This demonstrates that SpikeInit is also effective in large-scale datasets.

### 6.3. Machine Translation

Text data is inherently variable in length, i.e., sentences often have different lengths, whereas BatchNorm relies on a consistent, dense structure to compute a meaningful mean and variance. As a result, language transformers commonly adopt Layer Normalization (LN) instead of Batch Normalization (BN). However, LN introduces normalization operations during inference that require floating-point multiplications, which disrupt the spike-driven nature of SNNs. Therefore, neither BN nor LN is well-suited for spiking language transformers. To demonstrate the generality of SpikeInit and evaluate whether LN can be effectively removed from spiking language transformers by proper initialization, we conduct experiments on the IWSLT17 German-English machine translation task. We use a lightweight spik-

ing transformer (4-256). For comparison, we also trained the same architecture with either LN or BN under identical hyperparameter settings. The experimental results are listed in Tab. 3. SpikeInit without normalization outperforms both BN and LN, demonstrating that SpikeInit is also effective in replacing LN for natural language processing tasks, thereby expanding the applicability of deep SNNs to scenarios where BN is unsuitable.

### 6.4. Ablation Study

We perform an ablation study on key components of SpikeInit, including the weight initialization, the initialization of the shape parameter in the surrogate gradient function, and the adaptive surrogate gradient. Experiments are performed on CIFAR-10 using VGG-11 and MS ResNet-18 architectures, with hyperparameter settings following Section 6.2. The experimental results are listed in Tab. 4. We observe that a proper weight initialization is critical for training normalization-free deep SNNs. Without the weight initialization method introduced in Section 4, training often fails to converge. Additionally, it is also important to initialize the shape parameter of the surrogate gradient function. We observed a significant performance degradation on VGG-11 in the absence of surrogate gradient initialization. This is because an inappropriate initialization of surrogate gradient typically leads to vanishing or exploding gradients, making deep networks difficult to train effectively. For residual networks, this degradation was mitigated with the aid of residual connections. Residual blocks enable identity mapping at initialization, allowing gradients to propagate through blocks with a constant magnitude. However, issues of vanishing or exploding gradients may persist within individual blocks and between different network stages, leading to performance degradation. Based on the initialization of weights and surrogate gradients, the adaptive surrogate gradient helps further improve the performance. These findings demonstrate the effectiveness of each component.

### 7. Conclusion

In this work, we study how to train deep SNNs reliably without normalization by proper initialization. In Section 4, we model the responding curve of typical spiking layers, and develop a weight initialization method that maintains

stable firing rates across layers during forward propagation. In Section 5, we further formulate the gradient of spiking layers and initialize the shape parameter of the surrogate gradient function to ensure a stable gradient magnitude across layers during backpropagation. Combining these two methods, we propose SpikeInit, an initialization framework for SNNs. Extensive experiments on various datasets and network architectures demonstrate that SpikeInit can stably train deep SNNs without normalization and achieves better performance compared to normalized counterparts. Our work paves the way for large-scale normalization-free SNNs, liberating SNN design and applications from the constraints of normalization.

## Acknowledgements

This work is supported by STI 2030-Major Projects (2021ZD0200300), the National Natural Science Foundation of China (U24B20140, 62422601), Beijing Municipal Science and Technology Program (Z251100008125052), and Qiyuan Innovative Talent Program.

## Impact Statement

This paper presents work whose goal is to advance the field of Machine Learning. There are many potential societal consequences of our work, none which we feel must be specifically highlighted here.

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

# A. Proof of Theorems

## A.1. Proof of Theorem 4.1

*Proof.* Let $v$ denote the membrane potential following the stationary distribution, i.e., $v \sim \mathcal{N}(\mu_v, \sigma_v^2)$. The stationarity condition implies that the moments of $v[t+1]$ equal the moments of $v[t]$.

$$E[v[t+1]] = E[v[t]] = \mu_v, \tag{28}$$

$$E[v^2[t+1]] = E[v^2[t]] = \mu_v^2 + \sigma_v^2. \tag{29}$$

We first derive the equation of expectation. Take the expectation of both sides of Eq. (16):

$$E[v[t+1]] = \int_{-\infty}^{v_{\text{th}}} (\kappa u + E[x]) f_v(u) du + \int_{v_{\text{th}}}^{\infty} E[x] f_v(u) du. \tag{30}$$

Since $E[x] = 0$, the second integral (reset condition) vanishes, and $E[x]$ vanishes from the first term:

$$\mu_v = \kappa \int_{-\infty}^{v_{\text{th}}} u \frac{1}{\sigma_v} \phi\left(\frac{u - \mu_v}{\sigma_v}\right) du. \tag{31}$$

Let $z = \frac{u - \mu_v}{\sigma_v}$. Then $u = \sigma_v z + \mu_v$. The limit $v_{\text{th}}$ becomes $a = \frac{v_{\text{th}} - \mu_v}{\sigma_v}$.

$$\mu_v = \kappa \int_{-\infty}^{a} (\sigma_v z + \mu_v) \phi(z) dz. \tag{32}$$

Using the property of the truncated standard normal, $\int_{-\infty}^{a} z\phi(z) dz = -\phi(a)$:

$$\mu_v = \kappa(\sigma_v(-\phi(a)) + \mu_v \Phi(a)). \tag{33}$$

Rearranging to isolate terms yields the equation of expectation:

$$\mu_v(1 - \kappa\Phi(a)) = -\kappa\sigma_v\phi(a). \tag{34}$$

We then derive the equation for the second moment. Square both sides of Eq. (16):

$$v^2[t+1] = \begin{cases} \kappa^2 v^2[t] + 2\kappa v[t]x[t] + x^2[t] & \text{if } v[t] < v_{\text{th}} \\ x^2[t] & \text{if } v[t] \geq v_{\text{th}}. \end{cases} \tag{35}$$

Take the expectation:

$$E[v^2[t+1]] = \int_{-\infty}^{v_{\text{th}}} (\kappa^2 u^2 + \sigma_x^2) f_v(u) du + \int_{v_{\text{th}}}^{\infty} \sigma_x^2 f_v(u) du. \tag{36}$$

Note that the cross-term $E[2\kappa v[t]x[t]]$ is zero because $x[t]$ is independent of $v[t]$ and has mean zero. Factoring out $\sigma_x^2$, which integrates over the full PDF to 1:

$$\mu_v^2 + \sigma_v^2 = \sigma_x^2 + \kappa^2 \int_{-\infty}^{v_{\text{th}}} u^2 f_v(u) du. \tag{37}$$

Consider the integral $\mathcal{I}(v_{\text{th}}) = \int_{-\infty}^{v_{\text{th}}} u^2 f_v(u) du$. Substitute $u = \sigma_v z + \mu_v$:

$$\mathcal{I}(v_{\text{th}}) = \int_{-\infty}^{a} (\sigma_v^2 z^2 + 2\mu_v\sigma_v z + \mu_v^2)\phi(z) dz. \tag{38}$$

Using the standard identities:

$$\int_{-\infty}^{a} \phi(z) dz = \Phi(a), \tag{39}$$

$$\int_{-\infty}^{a} z\phi(z) dz = -\phi(a), \tag{40}$$

$$\int_{-\infty}^{a} z^2\phi(z) dz = \Phi(a) - a\phi(a). \tag{41}$$

We obtain:

$$\mathcal{I}(v_{\text{th}}) = \sigma_v^2(\Phi(a) - a\phi(a)) + 2\mu_v\sigma_v(-\phi(a)) + \mu_v^2\Phi(a)$$
$$= (\mu_v^2 + \sigma_v^2)\Phi(a) - \sigma_v(\mu_v + v_{\text{th}})\phi(a). \qquad (42)$$

Substituting $\mathcal{I}(v_{\text{th}})$ back into the stationarity equation yields the equation of the second moment. $\qquad\square$

### A.2. Proof of Theorem 5.1

*Proof.* Since the expectation of the gradient $E\left[\frac{\partial\mathcal{L}}{\partial s^{(l)}[t]}\right] = 0$, the variance of gradient $\text{Var}\left(\frac{\partial\mathcal{L}}{\partial s^{(l)}[t]}\right) = E\left[\left(\frac{\partial\mathcal{L}}{\partial s^{(l)}[t]}\right)^2\right]$. Let $G[i] = \frac{\partial\mathcal{L}}{\partial s^{(l+1)}[i]}$, $J[i] = \frac{\partial s^{(l+1)}[i]}{\partial v^{(l)}[i]}\frac{\partial v^{(l)}[i]}{\partial x^{(l)}[t]}W^{(l)}$, the square of gradient decompose to:

$$\left(\frac{\partial\mathcal{L}}{\partial s^{(l)}[t]}\right)^2 = \sum_{i=t}^{T}\sum_{j=t}^{T}G[i]J[i]G[j]J[j]. \qquad (43)$$

Take the expectation of both sides of the equation. Since the incoming gradients $G[i]$ are independent and zero-mean, all cross-terms vanish:

$$E[G[i]J[i]G[j]J[j]] = E[G[i]]E[G[j]]E[J[i]J[j]] = 0, i \neq j. \qquad (44)$$

When $i = j$, $E[G^2[i]J^2[i]] = \sigma_g^2E[J^2[i]]$. Since weights are independently drawn from a normal distribution with mean 0 and variance $\sigma_w^2$:

$$E[J^2[i]] = n_{\text{out}}^{(l)}\sigma_w^2 E\left[\left(\frac{\partial s^{(l+1)}[i]}{\partial v^{(l)}[i]}\frac{\partial v^{(l)}[i]}{\partial x^{(l)}[t]}\right)^2\right]. \qquad (45)$$

We obtain:

$$\text{Var}\left(\frac{\partial\mathcal{L}}{\partial s^{(l)}[t]}\right) = \sum_{i=t}^{T}n_{\text{out}}^{(l)}(\sigma_w^{(l)})^2\sigma_g^2 E\left[\left(\frac{\partial s^{(l+1)}[i]}{\partial v^{(l)}[i]}\frac{\partial v^{(l)}[i]}{\partial x^{(l)}[t]}\right)^2\right]. \qquad (46)$$

We evaluate the expected squared path term. The term decomposes into a decay factor and the squared derivative:

$$\left(\frac{\partial s^{(l+1)}[i]}{\partial v^{(l)}[i]}\frac{\partial v^{(l)}[i]}{\partial x^{(l)}[t]}\right)^2 = \left(\lambda\kappa^{i-t}\prod_{j=t}^{i}\mathbb{I}(v^{(l)}[j] < v_{\text{th}})\right)^2(\Theta'(v^{(l)} - v_{\text{th}};\alpha))^2, \qquad (47)$$

where $\mathbb{I}(\cdot)$ is the indicator function. According to the stationary distribution assumption and given firing rate $p$, we take the expectation and apply the independence approximation between the membrane potential accumulation and the derivative magnitude:

$$E\left[\left(\frac{\partial s^{(l+1)}[i]}{\partial v^{(l)}[i]}\frac{\partial v^{(l)}[i]}{\partial x^{(l)}[t]}\right)^2\right] \approx \lambda^2(\kappa^2(1-p))^{i-t}\cdot\mathcal{M}_2, \qquad (48)$$

where $\mathcal{M}_2 = E[(\Theta'(v^{(l)} - v_{\text{th}};\alpha))^2]$. Summing the terms yields a geometric series:

$$\text{Var}\left(\frac{\partial\mathcal{L}}{\partial s^{(l)}[t]}\right) \approx \sum_{i=t}^{T}n_{\text{out}}^{(l)}(\sigma_w^{(l)})^2\lambda^2\sigma_g^2\cdot\mathcal{M}_2\cdot(\kappa^2(1-p))^{i-t}. \qquad (49)$$

Assume that $T$ is large enough, the series converges to:

$$\text{Var}\left(\frac{\partial\mathcal{L}}{\partial s^{(l)}[t]}\right) \approx \frac{n_{\text{out}}^{(l)}(\sigma_w^{(l)})^2\lambda^2\sigma_g^2}{1 - \kappa^2(1-p)}\cdot\mathcal{M}_2. \qquad (50)$$

$\qquad\square$

### A.3. Derivation of Eq. (25)

**Theorem A.1.** *Let $v$ be a random variable following a normal distribution $v \sim \mathcal{N}(\mu_v, \sigma_v^2)$. Let $\Theta(v - v_{\text{th}}; \alpha)$ be a nonlinear activation function whose derivative is defined as:*

$$\Theta'(v - v_{\text{th}}; \alpha) = \alpha e^{-2\alpha|v - v_{\text{th}}|}, \tag{51}$$

*where $\alpha > 0$ is the shape parameter and $v_{\text{th}}$ is the threshold. Then, the expected value of the squared derivative, denoted $\mathcal{M}_2 = E[(\Theta'(v - v_{\text{th}}; \alpha))^2]$, is given by:*

$$\mathcal{M}_2 = \alpha^2 e^{8\alpha^2 \sigma_v^2} \left[ e^{4\alpha(\mu_v - v_{\text{th}})} \Phi\left(\frac{v_{\text{th}} - \mu_v - 4\alpha\sigma_v^2}{\sigma_v}\right) + e^{-4\alpha(\mu_v - v_{\text{th}})} \Phi\left(\frac{\mu_v - v_{\text{th}} - 4\alpha\sigma_v^2}{\sigma_v}\right) \right],$$

*where $\Phi(\cdot)$ is the CDF of the standard normal distribution.*

*Proof.* The squared derivative is $(\Theta'(v - v_{\text{th}}; \alpha))^2 = \alpha^2 e^{-4\alpha|v - v_{\text{th}}|}$. The expectation is defined as the integral against the PDF of $v$

$$\mathcal{M}_2 = \int_{-\infty}^{\infty} \alpha^2 e^{-4\alpha|v - v_{\text{th}}|} \frac{1}{\sqrt{2\pi}\sigma_v} e^{-\frac{(v - \mu_v)^2}{2\sigma_v^2}} dv. \tag{52}$$

We split the integral at the threshold $v_{\text{th}}$ to handle the absolute value term $|v - v_{\text{th}}|$:

$$\mathcal{M}_2 = \frac{\alpha^2}{\sqrt{2\pi}\sigma_v} \left[ \int_{-\infty}^{v_{\text{th}}} e^{4\alpha(v - v_{\text{th}})} e^{-\frac{(v - \mu_v)^2}{2\sigma_v^2}} dv + \int_{v_{\text{th}}}^{\infty} e^{-4\alpha(v - v_{\text{th}})} e^{-\frac{(v - \mu_v)^2}{2\sigma_v^2}} dv \right]. \tag{53}$$

Consider the generic integral form $\mathcal{I} = \int e^{kv} e^{-\frac{(v - \mu)^2}{2\sigma^2}} dv$. The exponent converts to:

$$-\frac{(v - (\mu + k\sigma^2))^2}{2\sigma^2} + \mu k + \frac{k^2 \sigma^2}{2}. \tag{54}$$

The integral becomes a shifted Gaussian PDF with mean $\mu + k\sigma^2$ scaled by a constant factor $C = e^{\mu k + \frac{k^2 \sigma^2}{2}}$.

Therefore, For the first integral ($v < v_{\text{th}}$), $k = 4\alpha$. The scaling factor is $C_1 = e^{-4\alpha v_{\text{th}}} \cdot e^{4\alpha\mu_v + \frac{(4\alpha)^2 \sigma_v^2}{2}} = e^{4\alpha(\mu_v - v_{\text{th}}) + 8\alpha^2 \sigma_v^2}$. The Gaussian is $\Phi\left(\frac{v_{\text{th}} - (\mu_v + 4\alpha\sigma_v^2)}{\sigma_v}\right)$.

$$\mathcal{I}_1 = \alpha^2 e^{4\alpha(\mu_v - v_{\text{th}}) + 8\alpha^2 \sigma_v^2} \Phi\left(\frac{v_{\text{th}} - \mu_v - 4\alpha\sigma_v^2}{\sigma_v}\right). \tag{55}$$

For second integral ($v \geq v_{\text{th}}$), $k = -4\alpha$. The scaling factor is $C_2 = e^{4\alpha v_{\text{th}}} \cdot e^{-4\alpha\mu_v + \frac{(-4\alpha)^2 \sigma_v^2}{2}} = e^{-4\alpha(\mu_v - v_{\text{th}}) + 8\alpha^2 \sigma_v^2}$. The Gaussian is $1 - \Phi\left(\frac{v_{\text{th}} - (\mu_v - 4\alpha\sigma_v^2)}{\sigma_v}\right) = \Phi\left(\frac{\mu_v - v_{\text{th}} - 4\alpha\sigma_v^2}{\sigma_v}\right)$.

$$\mathcal{I}_2 = \alpha^2 e^{-4\alpha(\mu_v - v_{\text{th}}) + 8\alpha^2 \sigma_v^2} \Phi\left(\frac{\mu_v - v_{\text{th}} - 4\alpha\sigma_v^2}{\sigma_v}\right). \tag{56}$$

Summing the two terms, we obtain:

$$\mathcal{M}_2 = \alpha^2 e^{8\alpha^2 \sigma_v^2} \left[ e^{4\alpha(\mu_v - v_{\text{th}})} \Phi\left(\frac{v_{\text{th}} - \mu_v - 4\alpha\sigma_v^2}{\sigma_v}\right) + e^{-4\alpha(\mu_v - v_{\text{th}})} \Phi\left(\frac{\mu_v - v_{\text{th}} - 4\alpha\sigma_v^2}{\sigma_v}\right) \right]. \tag{57}$$

$\square$

## B. Initialization of Special Layers

**Direct coding layer.** Recent deep SNNs typically adopt direct coding, where the first layer serves as a coding layer to convert input data, such as pixel intensity, into spike trains based directly on their raw values. We assume the input data are

*Table 5.* Results on CIFAR-10 with SEW ResNet.

| Method | Normalization Free | Accuracy (%) | |
|---|---|---|---|
| | | SEW ResNet-18 | SEW ResNet-34 |
| fluctuation-driven (Rossbroich et al., 2022) | ✓ | 91.75 | 91.22 |
| Ding et al. (2025) initialization | ✓ | 91.47 | 90.42 |
| Micheli et al. (2025) initialization | ✓ | 91.41 | 90.90 |
| BNTT (Kim & Panda, 2021) | ✗ | 94.27 | 94.61 |
| tdBN (Zheng et al., 2021) | ✗ | 95.01 | 95.43 |
| TEBN (Duan et al., 2022) | ✗ | 95.18 | 95.58 |
| MPBN (Guo et al., 2023) | ✗ | 94.18 | 95.62 |
| **SpikeInit (Ours)** | ✓ | **95.31** | **95.71** |

normalized to have zero mean and unit variance. For temporal dynamic inputs, including neuromorphic data and sequential data, we assume that the inputs are mutually independent across time steps. Consequently, the variance of the input current becomes $n_{\mathrm{in}}^{(1)}(\sigma_w^{(1)})^2$, which equals an input firing rate of 1 in Eq. (14). Therefore, we can search for a $\sigma^*$ that satisfies $p_{\mathrm{init}} = q(1)$ following a procedure similar to Algorithm 2.

For static inputs such as images, the input current remains constant over time. This makes the process formulated in Eq. 16 to become deterministic once the input is sampled. The firing rate of a static direct coding layer can be formulated as

$$p = \sum_{i=1}^{T} \frac{1}{i} \left[ \Phi\left( \frac{v_{\mathrm{th}}}{\sigma_x(1 - \kappa^{i-1})} \right) - \Phi\left( \frac{v_{\mathrm{th}}}{\sigma_x(1 - \kappa^{i})} \right) \right], \tag{58}$$

where $\sigma_x = \sqrt{n_{\mathrm{in}}^{(1)}}\lambda\sigma_w^{(1)}$, and with the convention that $1 - \kappa^0 = 0$, $\frac{v_{\mathrm{th}}}{\sigma_x(1-\kappa^0)} = \infty$, and $\Phi(\infty) = 1$. Therefore, we calculate $\sigma_w^{(1)}$ by solving the nonlinear equation Eq. (58) with $p = p_{\mathrm{init}}$.

**Residual blocks.** Residual learning and shortcuts have been proven to be an important method for training deep SNNs. Two types of residual connections are commonly used in residual SNNs: spike-element-wise (SEW) residual connection (Fang et al., 2021a) and membrane-shortcut (MS) residual connection (Hu et al., 2024), respectively represented as

$$S^{(l+1)}[t] = \mathrm{SN}(\mathcal{F}^{(l)}(S^{(l)}[t])) + S^{(l)}[t], \tag{59}$$

$$I^{(l+1)}[t] = \mathcal{F}^{(l)}(\mathrm{SN}(I^{(l)}[t])) + I^{(l)}[t], \tag{60}$$

where $\mathcal{F}^{(l)}(\cdot)$ represents the functions in the residual connection, $\mathrm{SN}(\cdot)$ represents the spiking neuron layer, $S$ and $I$ represent spikes and input currents, respectively.

For the MS connection, inspired by the initialization methods for residual branches in ANNs (Zhang et al., 2019), we initialize the last layer of each residual branch to 0 for the MS connection. This ensures the dynamical isometry for the initial state. However, for the SEW connection, zero-initializing the last layer of the residual branch leads to gradient vanishing. This is because the gradient of the residual branch is given by surrogate gradient $\partial S^{(l+1)}/\partial \mathcal{F}^{(l)} = \Theta'(\mathcal{F}^{(l)}(S^{(l)}))$. If the residual branch is initialized to 0, the output spiking neuron receives zero input current, causing the surrogate gradient to vanish. To solve this problem, we initialize the last layer of each residual branch in the SEW connection such that the output spiking neuron fires at a low firing rate $\epsilon$, typically set to 0.001. Since the firing rate is close to 0, we approximate the membrane potential as having zero mean, and the maximum variance is

$$\sum_{t=1}^{T} n_{\mathrm{in}} p_{\mathrm{max}} \lambda^2 \sigma_{\mathrm{SEW}}^2 \kappa^{t-1} \approx n_{\mathrm{in}} p_{\mathrm{max}} \lambda^2 \sigma_{\mathrm{SEW}}^2 \cdot \frac{1}{1 - \kappa}, \tag{61}$$

where $p_{\mathrm{max}}$ is the maximum input firing rate, typically set to 0.5. Therefore, we calculate $\sigma_{\mathrm{SEW}}$ by

$$\sigma_{\mathrm{SEW}} = \frac{v_{\mathrm{th}}(1 - k)}{\Phi^{-1}(1 - \epsilon)\sqrt{n_{\mathrm{in}} p_{\mathrm{max}}}\lambda}. \tag{62}$$

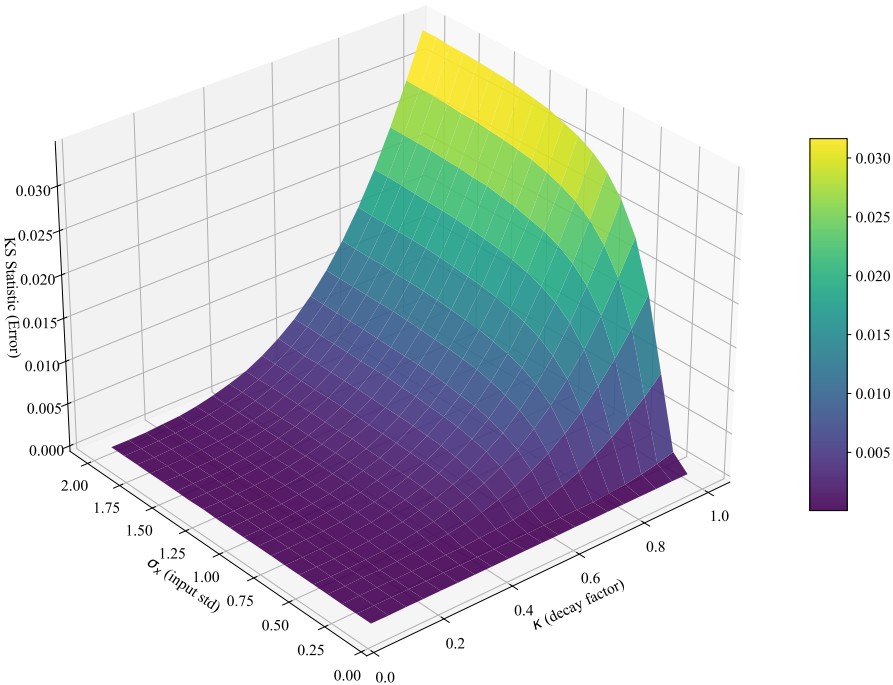

*Figure 4.* KS statistics vs decay factor $\kappa$ and standard deviation of input current $\sigma_x$.

In the main text, we use MS connection by default. We also conduct experiments using SEW ResNet on CIFAR-10. Hyperparameter settings follow the MS ResNet. Experimental results are listed in Tab. 5. SpikeInit also outperforms state-of-the-art methods when applied to SEW ResNet, demonstrating its generalizability across diverse architectures.

## C. Error Analysis and Simulation-based Initialization

### C.1. Error Analysis

In this section, we conduct an error analysis of the normal distribution approximation for the membrane potential. We compare the approximate normal distribution of the membrane potential with the simulated distribution obtained using the Monte Carlo method. The Kolmogorov–Smirnov (KS) statistic is employed as the evaluation metric. We calculate the KS statistic of the approximation under each decay factor $\kappa$ and standard deviation of input current $\sigma_x$. Results are demonstrated in Fig. 4. It is observed that the error is primarily influenced by the decay factor $\kappa$, while the effect of the input current is relatively minor. The KS statistics remain low for small values of $\kappa$, indicating that the approximate distribution aligns well with the actual distribution when $\kappa$ is small. As $\kappa$ approaches 1, the KS statistic increases sharply. At this point, the approximate distribution deviates from the actual distribution, suggesting that the approximation is inaccurate when $\kappa$ is close to 1.

### C.2. Simulation-based Initialization

Due to the inaccuracy of the normal distribution approximation when $\kappa$ is close to 1, we use a Monte Carlo simulation to replace Algorithm 1. Specifically, we sample $N \times T$ input currents from a normal distribution with mean 0 and variance $\lambda p^{(l)} \sigma_w^2$, simulate the neural dynamics of membrane potential for $T$ time steps, and calculate $p^{(l+1)}$. The process is summarized as Algorithm 3. For each simulation, we set $N = 10^6$. Then, we can perform a similar binary search as Algorithm 2 to calculate the $\sigma^*$, with Algorithm 1 replaced by Algorithm 3.

For the initialization of the shape parameter $\alpha$, we directly calculate $\mathcal{M}_2$ in Eq. (24) using the membrane potential samples

---

**Algorithm 3** Firing Rate Response (Simulation)

---

**Input:** $p^{(l)}, \sigma_w^2, \kappa, \lambda, v_{\text{th}}, N, T$
**Output:** $p^{(l+1)}$
1:   $\sigma_x \leftarrow \sqrt{\lambda p^{(l)} \sigma_w^2}$
2: **for** $i = 1, 2, \ldots, N$ **do**
3:     $v_i[0] \leftarrow 0$
4:     **for** $t = 1, 2, \ldots, T$ **do**
5:       Sample input current $x_i[t]$ from $\mathcal{N}(0, \sigma_x^2)$
6:       $v_i[t] \leftarrow \begin{cases} \kappa v_i[t-1] + x_i[t] & \text{if } v_i[t-1] < v_{\text{th}} \\ x_i[t] & \text{if } v_i[t-1] \geq v_{\text{th}} \end{cases}$
7:     **end for**
8: **end for**
9:   $p^{(l+1)} \leftarrow \frac{1}{NT} \sum_{i=1}^{N} \sum_{t=1}^{T} \mathbb{I}(v_i[t] \geq v_{\text{th}})$

---

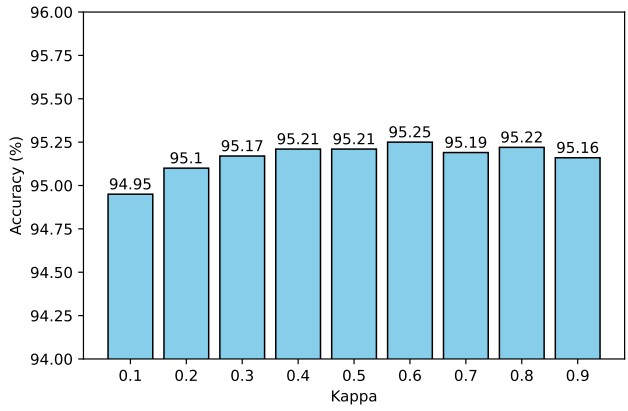

*Figure 5.* Results on CIFAR-10 with VGG-11 under different $\kappa$.

obtained from simulations via the law of large numbers

$$\hat{\mathcal{M}}_2 = \frac{1}{N} \sum_{i=1}^{N} (\Theta'(v_i - v_{\text{th}}; \alpha))^2 = \frac{1}{N} \sum_{i=1}^{N} \alpha^2 e^{-4|v_i - v_{\text{th}}|}. \tag{63}$$

Moreover, since $\kappa$ is close to 1, the series in Eq. (49) converges relatively slowly. Therefore, we use the actual value of the series rather than its limit:

$$\text{Var}\left(\frac{\partial \mathcal{L}}{\partial s^{(l)}[t]}\right) \approx n_{\text{out}}^{(l)} (\sigma_w^{(l)})^2 \lambda^2 \sigma_g^2 \cdot \hat{\mathcal{M}}_2 \frac{1 - (\kappa^2(1-p))^{T-t+1}}{1 - \kappa^2(1-p)}. \tag{64}$$

We take the average across time steps:

$$\frac{1}{T} \sum_{t=1}^{T} n_{\text{out}}^{(l)} (\sigma_w^{(l)})^2 \lambda^2 \sigma_g^2 \cdot \hat{\mathcal{M}}_2 \frac{1 - (\kappa^2(1-p))^{T-t+1}}{1 - \kappa^2(1-p)} = \hat{\mathcal{M}}_2 \frac{n_{\text{out}}^{(l)} (\sigma_w^{(l)})^2 \lambda^2 \sigma_g^2}{1 - \kappa^2(1-p)} \left(1 - \frac{1 - (\kappa^2(1-p))^T}{T(1 - \kappa^2(1-p))} \kappa^2(1-p)\right). \tag{65}$$

The function Eq. (24) becomes:

$$\hat{\mathcal{M}}_2 \left(1 - \frac{1 - (\kappa^2(1-p))^T}{T(1 - \kappa^2(1-p))} \kappa^2(1-p)\right) = \frac{1 - \kappa^2(1-p)}{n_{\text{out}}^{(l)} (\sigma_w^{(l)})^2 \lambda^2}. \tag{66}$$

Then we solve Eq. (66) for $\alpha^*$ using numerical methods.

*Table 6.* Detailed results on CIFAR-10 with different architectures. Accuracies are reported in the form of mean±std. - denotes the training fails to converge.

| Method | Spiking | Norm. Free | VGG-11 | | VGG-16 | | ResNet-18 | | ResNet-34 | |
| --- | --- | --- | --- | --- | --- | --- | --- | --- | --- | --- |
| | | | Acc. (%) | OPs (G) | Acc. (%) | OPs (G) | Acc. (%) | OPs (G) | Acc. (%) | OPs (G) |
| ANN w/ BN | ✗ | ✗ | 96.36±0.13 | 0.61 | 96.88±0.14 | 1.14 | 96.69±0.11 | 0.56 | 97.13±0.07 | 1.16 |
| ANN w/o BN | ✗ | ✓ | 96.05±0.11 | 0.61 | 96.64±0.10 | 1.14 | 96.47±0.07 | 0.56 | 96.86±0.11 | 1.16 |
| fluctuation-driven | ✓ | ✓ | - | - | - | - | 89.15±0.30 | 0.13 | 90.96±0.18 | 0.24 |
| Ding et al. init. | ✓ | ✓ | - | - | - | - | 89.57±0.12 | 0.13 | 91.16±0.04 | 0.25 |
| Micheli et al. init. | ✓ | ✓ | - | - | - | - | 90.50±0.10 | 0.15 | 91.83±0.07 | 0.29 |
| BNTT | ✓ | ✗ | 93.32±0.17 | 0.27 | 94.49±0.04 | 0.53 | 94.63±0.16 | 0.32 | 95.38±0.06 | 0.78 |
| tdBN | ✓ | ✗ | 93.56±0.20 | 0.17 | 92.86±0.44 | 0.33 | 95.37±0.08 | 0.24 | 95.86±0.17 | 0.52 |
| TEBN | ✓ | ✗ | 94.99±0.08 | 0.29 | 94.58±0.48 | 0.60 | 95.33±0.11 | 0.28 | 95.87±0.19 | 0.51 |
| MPBN | ✓ | ✗ | 94.62±0.13 | 0.26 | 95.31±0.07 | 0.67 | 94.64±0.25 | 0.32 | 95.47±0.02 | 0.82 |
| **SpikeInit (Ours)** | ✓ | ✓ | **95.29**±0.18 | **0.17** | **95.77**±0.09 | **0.33** | **95.53**±0.08 | **0.20** | **96.20**±0.10 | **0.34** |

*Table 7.* Detailed results on ImageNet with ResNet-34. - denotes the training fails to converge. ∗ denotes that this metric was not reported in the original paper.

| Method | Spiking | Normalization Free | T | Accuracy (%) | OPs (G) |
| --- | --- | --- | --- | --- | --- |
| ANN w/ BN | ✗ | ✗ | 1 | 71.51 | 3.68 |
| ANN w/o BN | ✗ | ✓ | 1 | - | - |
| tdBN (Zheng et al., 2021) | ✓ | ✗ | 6 | 63.72 | ∗ |
| MPBN (Guo et al., 2023) | ✓ | ✗ | 4 | 64.71 | ∗ |
| TAB (Jiang et al., 2024) | ✓ | ✗ | 4 | 67.78 | ∗ |
| TEBN (Duan et al., 2022) | ✓ | ✗ | 4 | 68.28 | ∗ |
| **SpikeInit (Ours)** | ✓ | ✓ | **4** | **68.81** | **1.41** |
| | | | **6** | **69.76** | **2.08** |

## C.3. Performance under Different $\kappa$

To evaluate the effectiveness of the simulation-based initialization of SpikeInit across different values of $\kappa$, we conduct experiments on CIFAR-10 using the VGG-11 architecture. For every $\kappa$, we set $\lambda = 1$, $v_{\text{th}} = 1$, and $p_{\text{init}} = 0.15$. The experimental results are demonstrated in Fig. 5. The accuracies obtained under various $\kappa$ are largely comparable, demonstrating that the simulation-based SpikeInit is applicable across a wide range of $\kappa$ values. It is worth noting that a slight decline in accuracy occurs when $\kappa$ is small, e.g., $\kappa = 0.1$. This can be attributed to the fact that LIF neurons with small $\kappa$ approximate the Heaviside function while lacking sufficient temporal dynamics, which limits their representational capacity.

## D. Detailed Experimental Results

We list the detailed experimental results on CIFAR-10 in Tab. 6. Each experiment is conducted three times using different random seeds, and the accuracies are reported as mean±std. As shown in Tab. 6, the accuracies of SpikeInit outperform the BN baselines, confirming that SpikeInit contributes to enhanced performance.

For comparison, we also conduct experiments on ANNs using the same hyperparameter settings. The ANNs adopt network architectures similar to those of the SNNs, with spiking neurons replaced by ReLU activations. The ANNs are initialized using Kaiming initialization (He et al., 2015). As shown in Tab. 6, deep ANNs with proper initialization can be trained without normalization. However, their accuracies without normalization are slightly lower compared to their normalized counterparts. This highlights the advantage of SpikeInit, which removes the dependence of deep SNN training on batch normalization while further enhancing performance.

We also list the number of operations, which denotes the Synaptic Operations (SOPs) for SNNs and the Multiply Accumulate

*Table 8.* Detailed results on DVS-CIFAR10 and DVSGesture with VGG-9. Accuracies are reported in the form of mean±std. - denotes the training fails to converge.

| Method | Spiking | Normalization Free | DVS-CIFAR10 | | DVSGesture | |
|---|---|---|---|---|---|---|
| | | | Acc. (%) | OPs (G) | Acc. (%) | OPs (G) |
| ANN w/ BN | ✗ | ✗ | 86.97±0.12 | 13.62 | 98.38±0.20 | 13.62 |
| ANN w/o BN | ✗ | ✓ | 81.17±0.15 | 13.62 | 97.45±0.20 | 13.62 |
| fluctuation-driven (Rossbroich et al., 2022) | ✓ | ✓ | - | - | - | - |
| Ding et al. (2025) initialization | ✓ | ✓ | - | - | - | - |
| Micheli et al. (2025) initialization | ✓ | ✓ | 72.23±1.05 | 1.24 | 92.94±1.60 | 0.82 |
| BNTT (Kim & Panda, 2021) | ✓ | ✗ | 84.40±0.44 | 2.56 | 96.76±0.20 | 1.72 |
| tdBN (Zheng et al., 2021) | ✓ | ✗ | 83.77±0.31 | 0.82 | 96.18±0.35 | 0.97 |
| TEBN (Duan et al., 2022) | ✓ | ✗ | 84.87±0.25 | 1.03 | 96.76±0.20 | 1.65 |
| MPBN (Guo et al., 2023) | ✓ | ✗ | 81.73±2.27 | 2.85 | 96.76±0.40 | 2.46 |
| **SpikeInit (Ours)** | ✓ | ✓ | **84.93±0.55** | **0.77** | **97.34±0.20** | **0.75** |

operations (MACs) for ANNs. As shown in Tab. 6, the number of operations of SpikeInit is smaller than both the BN baselines and the ANNs, demonstrating that SpikeInit can also reduce the SOPs and has energy efficiency advantages over ANNs.

Similar to CIFAR-10, we list the detailed experimental results on ImageNet in Tab. 7. As shown in Tab. 7, the training of ANNs without BN on ImageNet fails to converge. This indicates that simple weight initialization cannot guarantee stable training on large-scale datasets without normalization. In contrast, SNNs initialized using SpikeInit can be trained stably on large-scale datasets without normalization, and outperform their normalized counterparts.

We also conduct experiments on neuromorphic datasets, i.e., the DVS-CIFAR10 (Li et al., 2017) and DVSGesture (Amir et al., 2017) dataset. Similar to CIFAR-10, each experiment is conducted three times using different random seeds. We use the VGG-9 architecture, which is also denoted as VGGSNN in previous work (Deng et al., 2022; Duan et al., 2022; Shi et al., 2024a). As shown in Tab. 8, SpikeInit also outperforms normalized counterparts on neuromorphic datasets. Moreover, the number of operations of SpikeInit is smaller than the BN baselines and significantly smaller than that of the ANNs, demonstrating the energy efficiency advantages of SpikeInit.

# E. Experimental Settings

## E.1. Models

**VGG architectures.** The original VGG nets are designed for ImageNet classification with the input size of $224 \times 224$. We modify it to fit the reduced input size of CIFAR-10. Specifically, we remove the first max pooling layer and reduce the FC-4096 layer to FC-2048. Moreover, to mitigate the impact of max pooling on firing rates, we replace the max pooling with strided convolution, which achieves a similar downsampling effect.

For the neuromorphic datasets, we use the VGG-9 architecture, which is also denoted as VGGSNN in previous works (Deng et al., 2022; Duan et al., 2022; Shi et al., 2024a). It replaces the max pooling with average pooling, and uses only one FC layer for the classification head.

**ResNet architectures.** In the main text, we use MS ResNet by default. Note that the original MS ResNet introduces floating-point multiplication in downsampling shortcuts, since the inputs to the downsampling shortcut blocks are not spike-activated. We insert a spiking neuron layer before each downsampling convolution to maintain the spike-driven characteristic. For the CIFAR-10 classification task, we replace the $7 \times 7$ convolution with max pooling with a single 3 convolution to fit the reduced input size of CIFAR-10. For the ImageNet classification task, we apply downsampling in the first stage to replace max pooling, mitigating the impact of max pooling on firing rates.

The deep MS ResNet architecture in Section 6.1 follows the depth analysis in (Hu et al., 2024), which is a deep and narrow architecture. It contains 3 stages, with the number of filter channels being 16, 32, and 64 for each stage. Each stage has $N$ basic blocks (2 convolution layers). Specifically, the 1000-layer deep MS ResNet comprises: an initial $3 \times 3$ convolution,

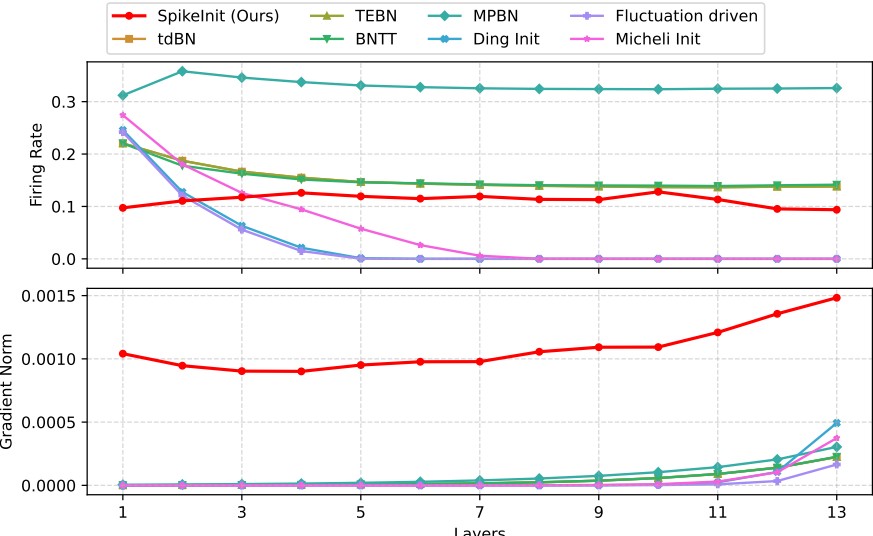

*Figure 6.* **Top**: Firing rate of each layer in VGG-16 (excluding fully connected layers) at initialization under different methods. SpikeInit achieves stable firing rates across layers that are lower than the normalized method, while firing rates of other initialization methods decay to 0 as network depth increases. **Bottom**: Gradient norm at initialization. SpikeInit achieves stable gradient norms across layers, while gradient norms of other methods decay to 0 during backpropagation.

166 blocks with 16 channels, 166 blocks with 32 channels, 167 blocks with 64 channels, and a fully-connected classifier layer, totaling 1000 layers. We set $v_{\text{th}} = 1$ for the deep MS ResNet.

**Transformer architectures.** We employ a lightweight spiking transformer for the machine translation task. It generally follows the vanilla Transformer architecture (encoder-decoder framework). We add spiking neuron layers before each linear transformation to replace the floating-point linear transformation. Both the encoder and decoder have 4 layers, with an embedding dimension of 256.

### E.2. Data Preprocessing and Hyperparameter Settings

**CIFAR-10 classification.** For the CIFAR-10 dataset, we perform data normalization to ensure that the input current of the direct coding layer has a mean of 0 and a variance of 1. We apply a random cropping to $32 \times 32$ with a padding of 4, random horizontal flipping, and auto-augment to avoid over-fitting.

For all experiments on CIFAR-10, we use 4 time steps. We use the SGD optimizer with a momentum of 0.9. One of the key advantages of batch normalization is that it allows training using a large learning rate. To evaluate whether this can also be achieved solely by proper initialization, we use a batch size of 256 and a learning rate of 0.1 with cosine decay to 0. Each model is trained for 300 epochs. We use a weight decay of $2 \times 10^{-4}$ to avoid over-fitting.

**ImageNet classification.** For the ImageNet dataset, we also perform data normalization. We apply standard data augmentation, i.e., random resized cropping to $224 \times 224$ and random horizontal flipping, to prevent overfitting.

Similar to the experiments on CIFAR-10, we use the SGDW optimizer with a momentum of 0.9, a batch size of 256, and a learning rate of 0.1 with cosine decay to 0. The model is trained for 300 epochs. We use a weight decay of $1 \times 10^{-4}$ to avoid over-fitting.

**IWSLT17 German-English translation.** For the IWSLT17 German-English machine translation task, we use 4 time steps. We use the AdamW optimizer with betas (0.9, 0.98). We use a batch size of 128 sentences and a learning rate of 0.001 with cosine decay to 0. The model is trained for 100 epochs. We use a weight decay of 0.01 and a dropout probability of 0.1 to avoid over-fitting.

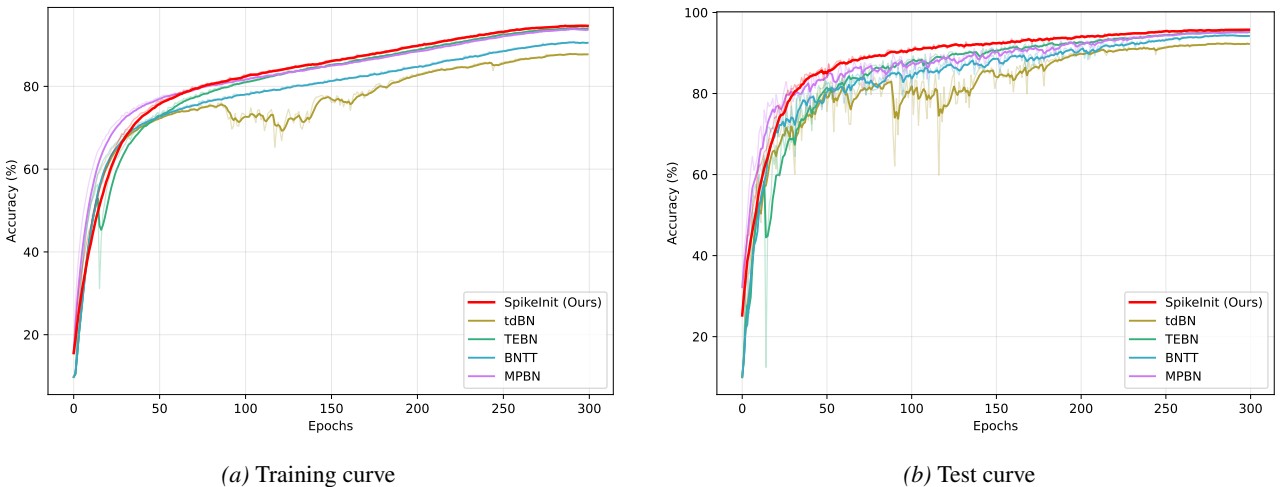

*(a)* Training curve                                                 *(b)* Test curve

*Figure 7.* Accuracy on CIFAR-10 training set and test set using VGG-16 under different methods during training.

**DVS data classification.**    For the DVS-CIFAR10 and DVSGesture datasets, we follow a specific preprocessing pipeline. First, we divide the event stream of each sample into 10 slices. Each slice corresponds to a continuous segment of the event stream, containing one-tenth of the total events in the sample. Next, for each slice, we compress all its events into a single 2-channel frame, with one channel for positive events and one for negative events. This frame is then used as the input for a single time step. As a result, the number of simulation steps used for the DVS datasets is 10. We resize the spatial size of the input to 48×48. We apply neuromorphic data augmentation (Li et al., 2022) and eventzoom (Dong et al., 2025) to prevent overfitting.

We use the SGD optimizer with a momentum of 0.9, a batch size of 64, and a learning rate of 0.025 with cosine decay to 0. The model is trained for 300 epochs on DVS-CIFAR10 and 600 epochs on DVSGesture. We use $\kappa = 0.25$ and a weight decay of $5 \times 10^{-4}$. We also apply the TET loss (Deng et al., 2022) for training.

## F. Firing Rates and Gradients at Initialization

To visually demonstrate that SpikeInit can maintain stable firing rates during forward propagation and stable gradient magnitudes during backpropagation at initialization, we conduct experiments to compute the firing rates and gradient norms of each spiking neuron layer in VGG-16 at initialization. We also compare the results with batch normalization methods (Zheng et al., 2021; Duan et al., 2022; Kim & Panda, 2021; Guo et al., 2023) and other initialization methods (Ding et al., 2025; Rossbroich et al., 2022; Micheli et al., 2025). The results are demonstrated in Fig. 6. During forward propagation at initialization, SpikeInit achieves a stable firing rate $p_{\text{init}}$ across layers. While normalization methods can also achieve stable firing rates by standardizing input currents or membrane potentials, their firing rates are generally higher than those of SpikeInit. In contrast, existing SNN initialization methods fail to keep a stable firing rate. The firing rates under existing SNN initialization methods decay to zero as network depth increases, leading to the training failing to converge. During backpropagation at initialization, SpikeInit maintains stable gradient norms across layers, whereas gradient norms of other methods decay to zero. These results highlight the effectiveness of SpikeInit in stabilizing both forward and backward propagation at initialization.

## G. Training Stability and Convergence

To visually demonstrate the training stability and convergence speed of SpikeInit, we plot the accuracy curve on both the training set and test set during training on CIFAR-10 using VGG-16. We also compare the results with those from batch normalization methods (Zheng et al., 2021; Duan et al., 2022; Kim & Panda, 2021; Guo et al., 2023). The results are demonstrated in Fig. 7. SpikeInit achieves a stable training process and exhibits convergence speed comparable to MPBN while outperforming BNTT. In contrast, the training processes of tdBN and TEBN are unstable, with the training curves showing significant fluctuations. Furthermore, the test accuracy curve of SpikeInit exhibits minimal fluctuation compared to that of normalization methods. This is primarily due to the inconsistency of Batch Normalization between training

and inference. During training, BN relies on statistics computed over mini-batches, whereas during inference it employs the exponential moving average of population statistics obtained in training. When parameter updates occur rapidly, the population statistics may deviate from the actual values, leading to performance fluctuations. These results demonstrate the superiority of SpikeInit.

## H. Computational Overhead

To evaluate the computational overhead of the initialization, we measured the time taken for initialization. We use AMD EPYC 7742 CPUs for the experiments. Results are listed in Tab. 9. The computation of SpikeInit can be done within one second. Even when using Monte Carlo simulations, it takes only a few seconds. This computational overhead is negligible compared to the hours-long training process.

*Table 9.* Computational overhead for initialization.

| Method | Time (s) |
| --- | --- |
| Weight initialization | 0.0215 |
| Surrogate gradient initialization | 0.186 |
| Weight initialization (Simulation) | 2.73 |
| Surrogate gradient initialization (Simulation) | 0.182 |

