# OpenReview forum: "Training Deep Spiking Neural Networks without Normalization"
_ICML.cc/2026/Conference — ICML 2026 regular_

### Official Review · Reviewer_TbCa · 2026-03-06

**Soundness:** 3
**Presentation:** 2
**Significance:** 3
**Originality:** 3
**Overall Recommendation:** 5
**Confidence:** 3

**Summary:**

The paper points out that training deep Spiking Neural Networks (SNNs) usually relies on Batch Normalization (BN) to stabilize input currents and gradients. However, BN has several limitations. For example, it depends on the batch size, is difficult to apply to variable-length sequence tasks (such as in NLP), and it disrupts the purely spike-based computation characteristic of SNNs. Therefore, the authors argue that the core issue is not that SNNs inherently require normalization, but rather that there is a lack of suitable initialization methods.

To address this problem, the paper proposes SpikeInit, an initialization framework designed for deep SNNs that enables stable training without using any normalization techniques (such as BatchNorm or LayerNorm). The method is based on theoretical analysis of signal and gradient propagation in SNNs. On one hand, it uses a firing rate response model to design weight initialization so that the spike firing rate of each layer remains stable during forward propagation. On the other hand, it initializes the shape parameter of the surrogate gradient and introduces an adaptive surrogate gradient, ensuring stable gradient propagation during backpropagation.

**Compliance With Llm Reviewing Policy:**

Affirmed.

**Ethical Review Concerns:**

- Is the assumption of a fixed FRR overly restrictive?
- Should Transformer-based models also be validated on vision-related tasks?
- The method jointly initializes weights and the surrogate gradient shape parameter. How sensitive is the training performance to the choice of the surrogate gradient parameters?

**Final Justification:**

The paper is well-structured and rigorous, and the authors have addressed all concerns during the rebuttal phase. Therefore, I have decided to raise my final score to 5.

**Key Questions For Authors:**

- Is the assumption of a fixed FRR overly restrictive?
- Should Transformer-based models also be validated on vision-related tasks?
- The method jointly initializes weights and the surrogate gradient shape parameter. How sensitive is the training performance to the choice of the surrogate gradient parameters?

**Limitations:**

see **Weaknesses**

**Strengths And Weaknesses:**

**Strengths**:
- The paper revisits the systematic limitations in SNN training and proposes an initialization and training mechanism for SNNs that does not rely on normalization.
- The method designs the initialization strategy based on the FRR (Firing Rate Response) model, making the firing rate during the forward pass more stable.
- SpikeInit, as an initialization strategy, does not require modifications to the network architecture and is simple to implement with low engineering cost.
- The method is theoretically compatible with various SNN architectures, giving it a certain level of generality.

**Weaknesses**:
- The firing rate response (FRR) is a statistical approximation and may not fully capture the complex temporal dynamics of real SNNs.
- Enforcing a fixed FRR might be overly restrictive for the model; it is worth considering whether allowing FRR to fluctuate within a reasonable range would be more appropriate.
- The shape parameter of the surrogate gradient and the adaptive mechanism may still require additional hyperparameter tuning.

---

> ### Author Rebuttal · Authors · 2026-03-30
>
> We appreciate the time and effort you invested in reviewing our paper. We will address all the questions you have raised.
>
> > Weaknesses 1: The firing rate response (FRR) is a statistical approximation and may not fully capture the complex temporal dynamics of real SNNs.
>
> We agree that the firing rate response (FRR) is a statistical approximation. However, for a randomly initialized SNN, its temporal dynamics are primarily determined by the statistical properties of the weights (mean and variance) and the hyperparameters of spiking neurons, as analyzed in Section 4.1. In the proposed weight initialization method, the temporal dynamics of the membrane potential have been formulated using a threshold auto-regressive (TAR) process, thereby fully accounting for the temporal dynamics of a randomly initialized SNN.
>
> For verification of this claim, please refer to our response to Reviewer sk3E Weakness 1. The weak temporal correlation under random initialization supports the reasonableness of analyzing the temporal dynamics of random initialized SNNs from the perspective of statistical approximation.
>
> > Question 1 and Weakness 2: Enforcing a fixed FRR might be overly restrictive for the model.
>
> The core principle of weight initialization methods is to maintain magnitude stability in activation values across network layers, i.e., $\mathrm{Var}(\mathrm{x}^{(l+1)})=\mathrm{Var}(\mathrm{x}^{(l)})$, through appropriate weight initialization. Based on this, the fixed FRR ensures that the network forward propagates at a stable firing rate $p_{\rm init}$ in its initial state, thereby preserving activation stability. If the FRR deviates from the input firing rate, the firing rate may either explode or vanish exponentially during forward propagation, leading to the training failing to converge.
>
> It is worth noting that the fixed FRR applies only at initialization. No constraints are imposed on the FRR during training. Once training begins, the network learns new spike response patterns, and the FRR and spike response patterns ultimately learned typically differ significantly from the initialization. Therefore, a fixed FRR at initialization does not limit the representational capacity of the network.
>
> > Question 2: Should Transformer-based models also be validated on vision-related tasks?
>
> Thank you for your suggestion. We add experiments on CIFAR-10 using QKFormer [1], a state-of-the-art spiking vision transformer architecture. Results are listed in Table R4.
>
> *Table R4. Results on CIFAR-10 using QKFormer.*
>
> | Method                    | Accuracy (%) |
> | ------------------------- | ------------ |
> | QKFormer (with tdBN)      | 96.18        |
> | QKFormer (with SpikeInit) | 96.26        |
>
> As shown in Table R4, SpikeInit outperforms the tdBN baseline, demonstrating that SpikeInit is also effective for spiking vision transformer.
>
> [1] Zhou, C., et al., QKFormer: Hierarchical Spiking Transformer using QK Attention. Advances in Neural Information Processing Systems, 2024.
>
> > Question 3 and Weakness 3: The shape parameter of the surrogate gradient and the adaptive mechanism may still require additional hyperparameter tuning. How sensitive is the training performance to the choice of the surrogate gradient parameters?
>
> By solving Equation (24), the proposed surrogate gradient initialization method automatically determines the optimal shape parameter $\alpha^*$ of the surrogate gradient. This ensures that the network maintains stable gradient magnitudes across layers during backpropagation at initialization, eliminating the need for additional hyperparameter tuning.
>
> To assess the sensitivity of training performance to the surrogate gradient parameters, we conduct experiments on CIFAR-10 using VGG-16 with perturbed values of $\alpha^*$. To focus on the effects of surrogate gradient parameters, we omit the adaptive surrogate gradient. Results are listed in Table R5.
>
> *Table R5. Results on CIFAR-10 with perturbed* $\alpha^*$.
>
> | Shape parameter     | Accuracy (%) |
> | ------------------- | ------------ |
> | $\alpha^*/2.0$      | 81.13        |
> | $\alpha^*/1.5$      | 92.06        |
> | $\alpha^*$          | 95.07        |
> | $\alpha^*\times1.5$ | 94.17        |
> | $\alpha^*\times2.0$ | 86.06        |
>
> As shown in Table R5, excessive deviation of the shape parameter from $\alpha^*$ leads to significant performance degradation. This validates the effectiveness of the proposed surrogate gradient initialization.

---

> > ### Author Rebuttal · Reviewer_TbCa · 2026-04-02
> >
> > Thank you to the authors for the response. The authors have addressed all the concerns well, and therefore I have decided to increase my score to 5.

---

### Official Review · Reviewer_1dyY · 2026-03-11

**Soundness:** 3
**Presentation:** 2
**Significance:** 4
**Originality:** 3
**Overall Recommendation:** 4
**Confidence:** 3

**Summary:**

- The authors propose SpikeInit, an initialization framework for SNNs without normalization via combinations of  stabling firing rats across layers during forward propagation as well as surrogating gradient during backpropagation.
- The core problem it addresses is the reliance of deep SNN training on Batch Normalization (BN), which has limitations in variable-length tasks and reduced batch sizes in conventional works.

**Compliance With Llm Reviewing Policy:**

Affirmed.

**Key Questions For Authors:**

- What was the main reason the performance without surrogate gradient initialization with VgG-11 in Table 4? Particularly, the ResNet-18 is more robust (while demonstrating the degradation of performance by few percents), raising the question of potential limitations of the proposed approach or unstableness in some architecture.

**Limitations:**

yes

**Strengths And Weaknesses:**

Strength:
- The proposed approach successfully trained ultra-deep (1000-layer) SNNs without normalization and achieving superior performance are very strong, which may open up new potential applications where BN is not suitable for SNNs.

Weakness
- While the proposed method shows promising results, the performance gains over existing normalization approaches appear relatively modest, given that all four baselines already achieve around 95% accuracy. It may help to further clarify the practical significance of this improvement and discuss scenarios where the proposed approach offers clear advantages.
- The effect of surrogate‑gradient initialization seems to vary across architectures, as reflected in Table 4. Providing additional interpretation or analysis of why different models respond differently could strengthen the discussion.
- Similarly, in Table 3, the improvement in BLEU score appears limited. Since BLEU is typically expressed on a 0–1 scale, a value of 29 (which I believe percentage) may appear modest.  It would be great to see why this change is meaningful in your context or offering additional qualitative or task‑specific evidence could help readers better appreciate the impact.

---

> ### Author Rebuttal · Authors · 2026-03-30
>
> We sincerely appreciate your helpful feedback. We will incorporate all your suggestions and answer your questions.
>
> > Question & Weakness 2: Performance degradation without surrogate gradient initialization on VGG-11.
>
> Thank you for pointing this out. When the surrogate gradient initialization proposed in Section 5 is removed, there is no guarantee that gradient magnitudes will remain stable across layers during backpropagation. This typically leads to vanishing or exploding gradients, making deep networks difficult to train effectively. As a result, a significant performance degradation is observed on VGG-11.
>
> In contrast, for residual networks, this degradation is mitigated due to the presence of residual connections. Residual blocks enable identity mapping at initialization, allowing gradients to propagate through blocks with a constant magnitude. However, issues of vanishing or exploding gradients may still persist within individual blocks and between different network stages, leading to performance degradation. This demonstrates the importance of the proposed surrogate gradient initialization.
>
> We will add this analysis to the final version.
>
> > Weaknesses 1 & 3: The performance gains over existing normalization approaches appear relatively modest. It may help to further clarify the practical significance of this improvement and discuss scenarios where the proposed approach offers clear advantages.
>
> Thank you for your suggestion. The experiments in Section 6.2 are designed to demonstrate that SpikeInit can achieve performance comparable to or even better than the BN baselines without relying on normalization. This indicates that normalization is not essential for training SNNs. With proper initialization, SNNs can perform competitively without normalization.
>
> Beyond the performance improvements, the significance of SpikeInit lies in its ability to eliminate the reliance of deep SNN training on BN, thereby expanding the applicability of deep SNNs to scenarios where BN is unsuitable. For example, BN is generally not well-suited for text data due to variability in sequence lengths. As a result, language models commonly adopt LN instead of BN. However, LN introduces normalization operations during inference that require dense floating-point multiplications, which disrupt the sparse spike-driven nature of SNNs. Therefore, neither BN nor LN is ideal for spiking language models. In this context, SpikeInit addresses the incompatibility of existing normalization methods while achieving superior performance.
>
> The experiments in Section 6.3 are designed to validate this claim. Compared to LN, BN performs significantly worse on the machine translation task, confirming its unsuitability for this application. While LN performs better, it disrupts the spike-driven nature of SNNs and is therefore also unsuitable for spiking language transformers. In contrast, SpikeInit is not only more suitable for spiking language transformers but also achieves better performance, highlighting its practical advantages.
>
> We will add this analysis to the final version.

---

> > ### Author Rebuttal · Reviewer_1dyY · 2026-04-02
> >
> > Thank you for the response. The additional explanations provided by the authors effectively address the previously unclear aspects. I recommend incorporating this analysis into the final version. While I appreciate the clarification provided in the rebuttal by authors, I have kept the score unchanged as no additional analysis was presented.

---

### Official Review · Reviewer_k3iB · 2026-03-11

**Soundness:** 3
**Presentation:** 4
**Significance:** 3
**Originality:** 3
**Overall Recommendation:** 5
**Confidence:** 3

**Summary:**

Deep Spiking Neural Networks often rely on batch normalization to improve training. However, batch normalization is not always ideal for certain tasks, such as language modeling. Good weight initialization can instead be used to stabilize training. The authors estimate the expected firing rate of a spiking neuron, choose a target firing rate, and use this information to determine an appropriate weight variance. Furthermore, since the surrogate gradient used to approximate the Heaviside function, an exponential function in their case, also affects the gradient variance, they determine an initial value for its parameter $\alpha$.

The initialization strategy requires solving a nonlinear system, but this computation is only performed once during initialization. Additionally, the authors introduce a method to adapt the surrogate gradient parameter $\alpha$ during training to normalize the gradient variance. Empirical results show that the proposed initialization scheme can scale to networks with up to 1000 layers.

**Compliance With Llm Reviewing Policy:**

Affirmed.

**Final Justification:**

This paper increases the ability to train very deep SNNs and it was very well written. I had no questions that needed a rebuttal and I maintain my positive evaluation.

**Key Questions For Authors:**

I have no questions for the authors.

**Limitations:**

Yes

**Strengths And Weaknesses:**

This paper is clean and well-written; all the math seems sound and is easy to follow. While the idea of using weight initialization to improve the forward and backward variance is not new, the application of it to SNN is. Furthermore, their adaptation of the surrogate gradient parameter for the backward pass seems somewhat novel as well. Given that network depth is an important factor in NN performance, the ability to train deeper SNNs without requiring normalization layers could be practically useful.

---

> ### Author Rebuttal · Authors · 2026-03-30
>
> Thank you for your time and effort in reviewing our manuscript. We sincerely appreciate your helpful feedback and are also grateful for your positive recognition of our research and its potential contributions to the field.
>
> If you have any further questions or require additional clarification, please do not hesitate to let us know. We would be gald to provide further information.
>
> Thank you once again for your support and constructive input.

---

> > ### Author Rebuttal · Reviewer_k3iB · 2026-04-03
> >
> > Thank you.

---

### Official Review · Reviewer_sk3E · 2026-03-12

**Soundness:** 3
**Presentation:** 2
**Significance:** 3
**Originality:** 4
**Overall Recommendation:** 4
**Confidence:** 4

**Summary:**

This paper tackles the long-standing dependence of deep Spiking Neural Networks (SNNs) on Batch Normalization (BN) for stable training. The authors argue that BN is not fundamentally necessary but merely compensates for poor weight initialization in SNNs. They propose SpikeInit, an initialization framework that models the temporal dynamics of spiking neurons via a threshold auto-regressive (TAR) process, approximating the stationary membrane potential distribution as Gaussian. From this, they derive a weight initialization scheme (Algorithm 2, binary search for $\sigma_w^*$) to maintain stable firing rates across layers, and a surrogate gradient shape parameter initialization (via Theorem 5.1 and Eq. 25) to stabilize gradient magnitudes during backpropagation. An adaptive surrogate gradient mechanism further adjusts $\alpha$ during training based on sample-level membrane potential statistics. Experiments cover CIFAR-10 (VGG, ResNet), ImageNet (MS ResNet-34), and IWSLT17 machine translation with a spiking Transformer, demonstrating that SpikeInit matches or exceeds BN-based counterparts. A 1000-layer SNN is also trained successfully without normalization.

**Compliance With Llm Reviewing Policy:**

Affirmed.

**Key Questions For Authors:**

Several entries in Table 1 show improvements of 0.2%~0.5% over BN baselines. Could you provide standard deviations over multiple runs to confirm these are statistically significant rather than within noise? If improvements are not significant on CIFAR-10, I would reconsider the "superior performance" claim but the method could still be valuable for its qualitative benefits (no BN needed).

**Limitations:**

yes

**Strengths And Weaknesses:**

Strengths:
The theoretical analysis is more rigorous than prior SNN initialization works (Rossbroich et al., 2022; Ding et al., 2025; Micheli et al., 2025). Modeling the membrane potential as a TAR process and deriving its stationary distribution (Theorem 4.1) provides a principled foundation. The normal approximation is validated empirically with Monte Carlo simulations (Fig. 1), and the error analysis in Appendix D with KS statistics adds credibility. Both the forward pass (firing rate stability) and backward pass (gradient magnitude stability) are addressed, which is a more complete treatment than previous work that only considers one direction.

Removing the BN dependence in SNNs is a practically important problem. BN is incompatible with variable-length sequences and small batch sizes, which limits SNN applicability beyond fixed-size vision tasks. The NLP experiment (Table 3, spiking Transformer on IWSLT17) is particularly noteworthy as it demonstrates the practical benefit of normalization-free training in a domain where BN fundamentally cannot be used.

Weakness：
The Bernoulli independence assumption (Eq. 13) is a standard simplification, but spiking activity in trained networks exhibits strong temporal correlations due to shared inputs and recurrent dynamics. The paper does not discuss how violations of this assumption affect the initialization quality during actual training.

The gradient variance analysis (Theorem 5.1, Eq. 22) assumes independence between the membrane potential distribution and the surrogate gradient derivative. This is a strong assumption since both depend on the same input signals. The practical impact of this approximation gap is not quantified.

---

> ### Author Rebuttal · Authors · 2026-03-30
>
> We sincerely appreciate your thorough review of our paper and the invaluable insights you provided. We are grateful for the opportunity to address the points you raised.
>
> > Question: Standard deviations.
>
> Thanks for your suggestion. We have added the standard deviation for each data point in the form of  mean$\pm$std and included it in Table R1.
>
> *Table R1. Results on CIFAR-10.*
>
> | Method              | VGG-11         | VGG-16         | ResNet-18      | ResNet-34      |
> | ------------------- | -------------- | -------------- | -------------- | -------------- |
> | fluctuation-driven  | -              | -              | 89.15$\pm$0.30 | 90.96$\pm$0.18 |
> | Ding et al. Init    | -              | -              | 89.57$\pm$0.12 | 91.16$\pm$0.04 |
> | Micheli et al. Init | -              | -              | 90.50$\pm$0.10 | 91.83$\pm$0.07 |
> | BNTT                | 93.32$\pm$0.17 | 94.49$\pm$0.04 | 94.63$\pm$0.16 | 95.38$\pm$0.06 |
> | tdBN                | 93.56$\pm$0.20 | 92.86$\pm$0.44 | 95.37$\pm$0.08 | 95.86$\pm$0.17 |
> | TEBN                | 94.99$\pm$0.08 | 94.58$\pm$0.48 | 95.33$\pm$0.11 | 95.87$\pm$0.19 |
> | MPBN                | 94.62$\pm$0.13 | 95.31$\pm$0.07 | 94.64$\pm$0.25 | 95.47$\pm$0.02 |
> | **SpikeInit**       | 95.29$\pm$0.18 | 95.77$\pm$0.09 | 95.53$\pm$0.08 | 96.20$\pm$0.10 |
>
> As shown in Table R1, the accuracy improvement of SpikeInit over the BN baselines exceeds the margin of random error, confirming that SpikeInit contributes to enhanced performance.
>
> > Weaknesses 1: Bernoulli independence assumption.
>
> Thank you for pointing this out. We agree that spiking activity in trained SNNs exhibits strong temporal correlations. However, in randomly initialized SNNs, such temporal correlation is weak and can generally be neglected. To verify this claim, we calculate the covariance matrix of the temporal distribution of the spike outputs from the final convolutional layer of VGG-16. The network is initialized using SpikeInit and has not been trained. Results are listed in Table R2.
>
> *Table R2. Covariance matrix of the temporal distribution.*
>
> | T    | 1                   | 2                   | 3                    | 4                    |
> | ---- | ------------------- | ------------------- | -------------------- | -------------------- |
> | 1    | $2.47\times10^{-3}$ | $2.78\times10^{-4}$ | $2.07\times10^{-4}$  | $1.02\times10^{-4}$  |
> | 2    | $2.78\times10^{-4}$ | $4.29\times10^{-2}$ | $1.03\times10^{-3}$  | $2.03\times10^{-3}$  |
> | 3    | $2.07\times10^{-4}$ | $1.03\times10^{-3}$ | $6.30\times10^{-2}$  | $-1.36\times10^{-3}$ |
> | 4    | $1.02\times10^{-4}$ | $2.03\times10^{-3}$ | $-1.36\times10^{-3}$ | $7.54\times10^{-2}$  |
>
> As shown in Table R2, the covariance is approximately one order of magnitude smaller than the variance, confirming that this temporal correlation is weak under random initialization.
>
> > Weaknesses 2: Gradient independence assumption.
>
> Thank you for pointing this out. We acknowledge that the gradient independence assumption is strong since both depend on the same input signals. However, under the assumption that the membrane potential follows a stationary distribution, we treat the membrane potential distribution and firing rate (transition probability) as known. This allows us to effectively decouple the surrogate gradient $\partial s[i]/\partial v[i]$ from the transition gradient $\partial v[i]/\partial x[j]$.
>
> Similar to Weakness 1, we calculate the covariance $Cov((\partial s[i]/\partial v[i])^2, (\partial v[i]/\partial x[j])^2),1\le j\le i\le T$ and the expectation $E((\partial s[i]/\partial v[i]\cdot\partial v[i]/\partial x[j])^2)$ from the final convolutional layer of VGG-16. Results are listed in Table R3 in the form of covariance/expectation.
>
> *Table R3. Covariance and expectation.*
>
> | i \ j | 1                          | 2                         | 3                         | 4        |
> | ----- | -------------------------- | ------------------------- | ------------------------- | -------- |
> | 1     | $0/0.92$                   | -                         | -                         | -        |
> | 2     | $1.11\times10^{-4}/0.14$   | $0/0.58$                  | -                         | -        |
> | 3     | $2.28\times10^{-4}/0.029$  | $8.57\times10^{-4}/0.12$  | $0/0.49$                  | -        |
> | 4     | $4.54\times10^{-5}/0.0059$ | $1.76\times10^{-4}/0.024$ | $1.85\times10^{-4}/0.097$ | $0/0.42$ |
>
> Note that when $i=j$, the covariance is zero because the transition probability is always equal to one in this case. As shown in Table R3, the covariance is significantly smaller than the expectation, indicating that it can be reasonably ignored.

---

### Decision · Program_Chairs · 2026-04-30

**Decision:**

Accept (regular)

**Comment:**

**Summary of reviews.** Four reviewers assess the paper positively (two Accept at 5, two Weak Accept at 4). The paper demonstrates that Batch Normalization — long considered essential for deep SNN training — can be entirely eliminated through principled weight initialization derived from a Threshold-affected Autoregressive (TAR) process model of membrane potentials. The method enables stable training of SNNs up to 1,000 layers deep and extends to domains where BN is fundamentally unsuitable (variable-length NLP sequences, spike-driven inference).

**Key strengths.**
- First principled demonstration that deep SNNs (up to 1,000 layers) can be stably trained without any normalization layers — a significant practical and conceptual advance (consensus).
- Theoretically rigorous: the TAR process model (Theorem 4.1) for membrane potential dynamics is more principled than prior SNN initialization works (Reviewer sk3E).
- Addresses both forward (firing rate stability) and backward (surrogate gradient magnitude stability) passes — more complete than prior methods that only handle one direction (Reviewer sk3E).
- Practically important: BN is incompatible with variable-length sequences and small batches. The spiking Transformer NLP experiment (IWSLT17, Table 3) directly demonstrates this benefit (Reviewer sk3E).
- No architectural modifications required; low engineering cost (Reviewer TbCa).

**Key weaknesses and rebuttal assessment.**
- Bernoulli independence assumption ignores temporal correlations (Reviewer sk3E) → authors showed off-diagonal covariances are ~10× smaller than diagonal variances under random initialization, validating the approximation at init time. **Addressed, though sk3E did not formally acknowledge.**
- Performance gains over BN baselines appear modest on CIFAR-10 (Reviewer 1dyY) → authors reframed: the contribution is not beating BN on CIFAR-10 but proving BN is *unnecessary*, enabling new application domains. Standard deviations confirm gains are statistically significant (e.g., 95.77 ± 0.09 vs. 95.31 ± 0.07 on VGG-16). **Adequately addressed.**
- Architecture-dependent effects: VGG-11 degrades severely without surrogate gradient initialization, while ResNet-18 is more robust (Reviewer 1dyY) → explained by residual connections shielding ResNets from gradient instability. **Resolved.**
- Sensitivity to surrogate gradient parameter (Reviewer TbCa) → perturbation experiments show 1.5–2× deviation from optimal alpha* causes accuracy drops from 95.07% to 81–86%, confirming the initialization is critical and well-calibrated. **Resolved (score raised).**
- Additional Transformer validation requested (Reviewer TbCa) → QKFormer on CIFAR-10 added: SpikeInit 96.26% vs. tdBN 96.18%. **Resolved.**

**Discussion outcome.** Reviewer k3iB (Accept, conf 3) had no concerns and praised the paper's writing and contribution. Reviewer TbCa explicitly raised their score after rebuttal. Reviewers sk3E and 1dyY marked concerns as resolved but kept scores at 4; sk3E had no formal acknowledgement, and 1dyY noted "no additional analysis was presented" (though the rebuttal did provide new data). The overall signal is positive (4, 4, 5, 5).